# The architecture of functional lateralisation and its relationship to callosal connectivity in the human brain

Vyacheslav R. Karolis [1,2], Maurizio Corbetta[3,4,5,6,7,8] & Michel Thiebaut de Schotten [1,2,9]

Functional lateralisation is a fundamental principle of the human brain. However, a comprehensive taxonomy of functional lateralisation and its organisation in the brain is missing. Here, we report the first complete map of functional hemispheric asymmetries in the human brain, reveal its low dimensional structure, and its relationship with structural interhemispheric connectivity. Our results suggest that the lateralisation of brain functions is distributed along four functional axes: symbolic communication, perception/action, emotion, and decision-making. The similarity between this finding and recent work on neurological symptoms give rise to new hypotheses on the mechanisms that support brain recovery after a brain lesion. We also report that cortical regions showing asymmetries in task-evoked activity have reduced connections with the opposite hemisphere. This latter result suggests that during evolution, brain size expansion led to functional lateralisation to avoid excessive conduction delays between the hemispheres.

[1] Brain Connectivity and Behaviour Laboratory, Sorbonne Universities, Paris, France. [2] Frontlab, Institut du Cerveau et de la Moelle épinière (ICM), Sorbonne Universities, Inserm U 1127, CNRS UMR 7225, Paris, France. [3] Department of Neuroscience, University of Padova, Padova, Italy. [4] Padova Neuroscience Center (PNC), University of Padova, Padova, Italy. [5] Venetian Institute of Molecular Medicine, Fondazione Biomedica, Padova, Italy. [6] Department of Neurology, Washington University, Saint Louis, MO, USA. [7] Department of Radiology, Washington University, Saint Louis, MO, USA. [8] Department of Neuroscience, Washington University, Saint Louis, MO, USA. [9] Groupe d'Imagerie Neurofonctionnelle, Institut des Maladies Neurodégénératives-UMR 5293, CNRS, CEA University of Bordeaux, Bordeaux, France. Correspondence and requests for materials should be addressed to V.R.K. (email: vyacheslav. karolis@ndcn.ox.ac.uk) or to M.T.d S. (email: michel.thiebaut@gmail.com)

"A" re you left- or right-brain?". The widespread belief that hemispheric dominance influences the human character comes from a misinterpretation of several decades of neuropsychological findings[1] that show that functional lateralisation is a fundamental principle of the brain's organisation[2–4]. Today, after nearly 30 years of functional neuroimaging, theories on functional lateralisation suggest a less radical division and assume that the two hemispheres balance one another[5]. However, despite the implications of functional lateralisation theories for neurodevelopmental and psychiatric disorders[6,7], as well as for stroke recovery[8–11], a comprehensive mapping of functional lateralisation in the brain is, to our knowledge, still missing in the literature. It is also not known whether putatively lateralised cognitive functions share similar or different spatial patterns of functional activation and whether these functional activations can be categorised to a limited number of spatial patterns—have a low-dimensional structure.

Furthermore, the mechanisms that sustain functional lateralisation, and related inter-hemispheric communication, remain debated[12,13]. Two competing hypotheses have been proposed on the emergence of functional lateralisation based on the structure of the corpus callosum, the most considerable inter-hemispheric connection. The inter-hemispheric independence hypothesis suggests that, during evolution, brain size expansion led to functional lateralisation in order to avoid excessive conduction delays between the hemispheres[14]. Accordingly, functionally lateralised regions will be connected less strongly via the corpus callosum than non-lateralised regions to make processing of lateralised functions more efficient[15]. The inter-hemispheric competition hypothesis proposes that functional lateralisation arises from the competition between the hemispheres that inhibit each other via the corpus callosum. As functionally lateralised regions would need to inhibit the opposite hemisphere more than non-lateralised regions, they could be more connected by the corpus callosum. Preliminary anatomical[16] and functional magnetic resonance imaging (fMRI)[17] studies provide support for both theories. However, the small range of functions investigated and shortcomings in the methods often limit the interpretability of the findings[13]. Overall, the generalisation of these theories and findings to the whole brain's functional organisation remains unknown.

Here, we took advantage of combining the largest fMRI meta-analytic dataset[18] with the highest quality structural connectivity data[19] to produce, for the first time, a comprehensive map of the functional brain architecture of lateralised cognitive functions, characterise its low-dimensional structure, and examine its relationship to corpus callosum connectivity.

## Results

**Functional lateralisation maps and their low-dimensional structure**. We selected 590 terms related to specific cognitive processes out of the whole Neurosynth database (see Supplementary Table 1). A functional lateralisation map was computed for each term by calculating the difference between hemispheres for each pair of homologous voxels. Homologous functional regions may be displaced in the two hemispheres because of anatomical factors, e.g. the Yakovlevian torque[4,20], the size of the planum temporale[21] and motor cortex[22,23]. Here we adjusted for main anatomical asymmetries in the two hemispheres by registering the maps to a symmetric atlas[24].

We first sought to determine which regions show a significant functional lateralisation. Given that selected terms could be either correlated or related in a trivial way (e.g., singular and plural forms of the same word; "visual form" and "visual forms"), a varimax-rotated principal component analysis was run in order to

eliminate redundancy in the data. One hundred and seventy-one principal components with eigenvalues higher than the grand average were retained, explaining 72.6% of the variance of the lateralisation maps. General linear modelling was subsequently employed with component loads as a set of predictors to fit lateralisation maps data and identify voxels with a significant lateralisation associated with each component. After 5000 permutations, 25 principal components showed voxels with a significant lateralisation (>20 voxels; $P < 0.05$ family-wise error corrected; see Supplementary Table 2). Essentially, these represent the significant groups of voxels showing significant functional lateralisation in Neurosynth.

Next, a multivariate spectral embedding, based on the similarity between lateralisation maps, enabled us to examine a generic structure of the brain's functional lateralisation profiles, i.e. its "morphospace"[25,26]. The preliminary step that included the embedding in the first two dimensions (Fig. 1a and Supplementary Figure 1) revealed a triangular organisation of the lateralisation maps with three vertices: symbolic communication, perception/action and emotion. A $t$-ratio test, i.e. a test of i.e. pareto optimality,[27], between the organisation of real data and 2000 samples of simulated data, which were obtained via permutations of the voxel order before computing right–left differences, confirmed the statistical veracity of such triangular organisation. The same analysis was used to explore other dimensions and revealed three additional triangles and a fourth vertex given by decision-making. (Fig. 1b and Supplementary Figure 2).

Furthermore, by regressing lateralisation profiles onto terms' coordinates in the embedded space, we constructed predictions for the maps located at the coordinates of the vertices, which we will refer to as archetype maps.

The archetype maps corresponding to the symbolic communication axis was characterised by a left dominant activation of the dorsal and ventral posterior part of the frontal lobe, including Broca area and the presupplementary motor area, the posterior part of the temporal lobe, including Wernicke area and the Visual Word Form Area (i.e. VWFA). Right dominant activations were located in the posterior lobe of the cerebellum, including area Crus II (Fig. 2a).

The archetype perception/action map involved left sensorimotor cortex, left SMA and left thalamus. Right dominant activations included frontal eye field, intraparietal region, and ventral frontal regions, frontal eye field, presupplementary motor area, basal forebrain and anterior cerebellum (i.e. Areas V/VI and VIII) as well as part of the vermis (Fig. 2b).

The archetype emotion map involved the left anterior cingulate cortex, the basolateral complex of the right amygdala, the posterior part of the right inferior frontal gyrus, the right intraparietal sulcus and the posterior part of the right temporal lobe (Fig. 2c).

Finally, the decision-making archetype map involved mostly the right prefrontal cortex (i.e. medial orbital gyrus), the right frontal eye field, the left intraparietal sulcus together with the striatum (right putamen and left caudate) and the left basal forebrain (Fig. 2d).

**Corpus callosum and functional lateralisation**. Given that the microscopic diffusion of water molecules in the brain is easier along rather than across axons, tractography derived from diffusion-weighted magnetic resonance imaging allows for peering into the structural organisation of brain connectivity (Fig. 3a).

In the following, we analysed the relationship between functional lateralisation and corpus callosum connectivity measures by contrasting the connectivity profiles of lateralised

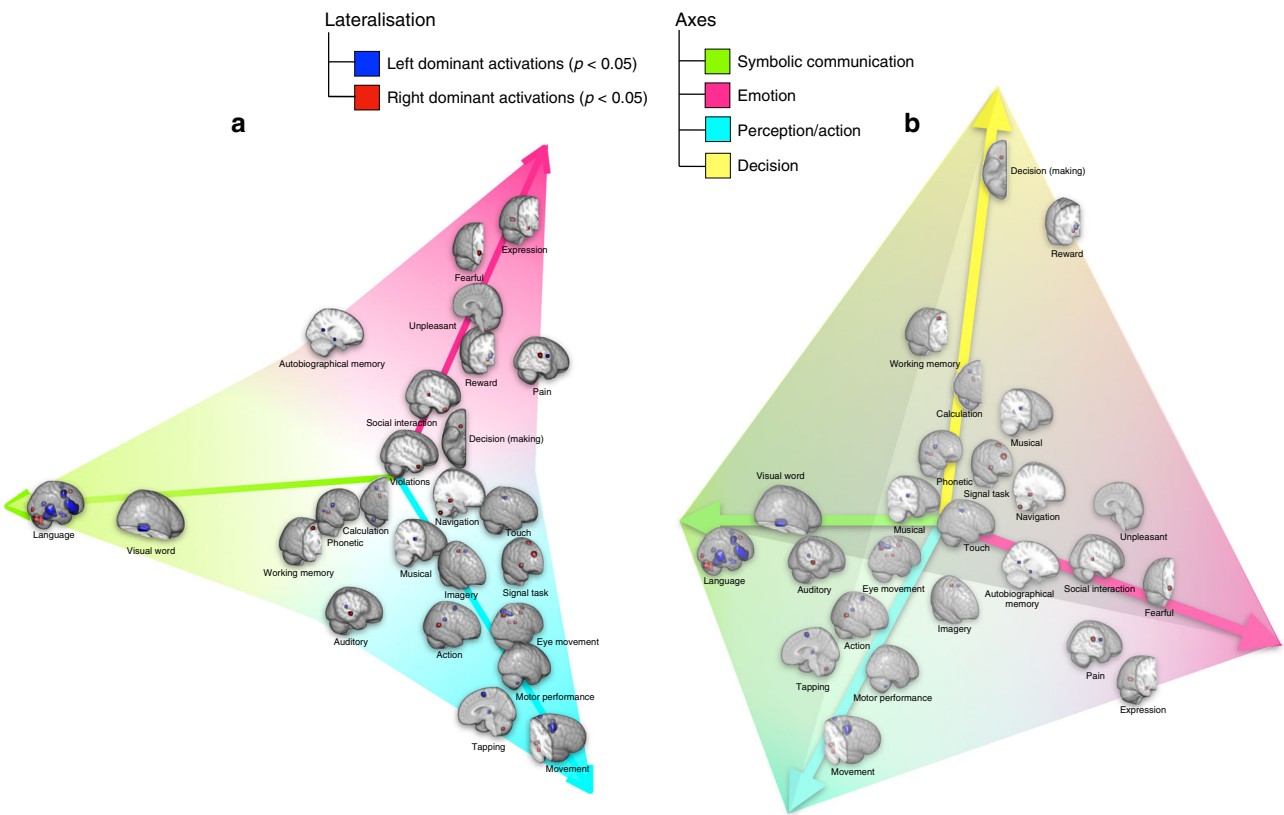

**Fig. 1** Low-dimensional structure of functional lateralisation. Embedded in two-dimensional (**a**) and three-dimensional (**b**) space according to similarity in their lateralisation profile (MATLAB interactive 3D file available as Supplementary Data 1). See Supplementary Figures 1 and 2 for the spatial embedding of all Neurosynth terms. Here, to provide a graphical summary of all results reported in the section, we plotted the significantly lateralised components maps, named by the highest-loading terms (Supplementary Table 3), in place of the actual Neurosynth terms (Supplementary Table 1)

and non-lateralised regions (see Methods for the definition of non-lateralised regions). Two measures of connectivity were employed, both computed by averaging across participants in the HCP sample. The first measure, axonal water fraction[28], is microscopic and is estimated in the midsection of the corpus callosum crossed by streamlines originating from voxels of a selected cortical region. The second measure is macroscopic and estimates the replicability of connections[29,30] calculated as the proportion of participants in which a voxel is connected to the corpus callosum, which we will refer to as probability of connection for shortness.

By sampling voxels from lateralised and non-lateralised regions, in each hemisphere separately, we first constructed the distribution of the differences in the axonal water fraction between lateralised and non-lateralised regions. Figure 3b indicates that the axonal water fraction was consistently lower for corpus callosum voxels projecting onto lateralised regions when compared to non-lateralised voxels. Additionally, the plots suggested a slightly lower axonal water fraction for left hemisphere regions as compared to the right hemisphere.

Next, we constructed an analogous distribution for the probability of connection. Figure 3c demonstrates that lateralised regions when compared to non-lateralised voxels did not differ in this macrostructural measure of connectivity.

The previous analysis failed to reveal a categorical difference between lateralised and non-lateralised regions in macroscopic measure of connectivity. However, the degree of functional hemispheric dominance (see Methods for the definition of this measure) can vary —from a unilateral to a relatively asymmetric pattern of activity. In the latter case, both hemispheres are

involved in a function, but one is more active than the other. Therefore, we explored whether a proportional relationship existed between the degree of functional lateralisation and the probability of corpus callosum connectivity.

Figure 3d indicates a negative relationship between the probability of connection and the degree of functional lateralisation, for both the left and the right hemispheres (Pearson correlation $r = -0.81$ and $r = -0.69$, respectively, $p < 0.001$). As the overall level of activation of two homotopic areas in the left and the right hemispheres may have an influence on its corpus callosum connections, we duplicated the same analysis after regressing out the left and right hemispheres average level of activity for every functionally lateralised voxel. The relationship between the level of functional dominance and the probability of connection to corpus callosum remained unchanged for the left hemisphere (Pearson correlation $r = -0.79$) and increased for the right hemisphere (Pearson correlation $r = -0.85$).

Additional supplementary analyses indicated that there was no relationship between the difference in corpus callosum connectivity of lateralised and non-lateralised voxels and their distance from the midsection of the corpus callosum (Supplementary Figure 3).

## Discussion

In the present study, we provide for the first time a comprehensive mapping of the functional brain architecture of lateralised cognitive functions. The lateralisation of brain functions had a low-dimensional structure distributed along four functional axes: symbolic communication, perception/action, emotion and

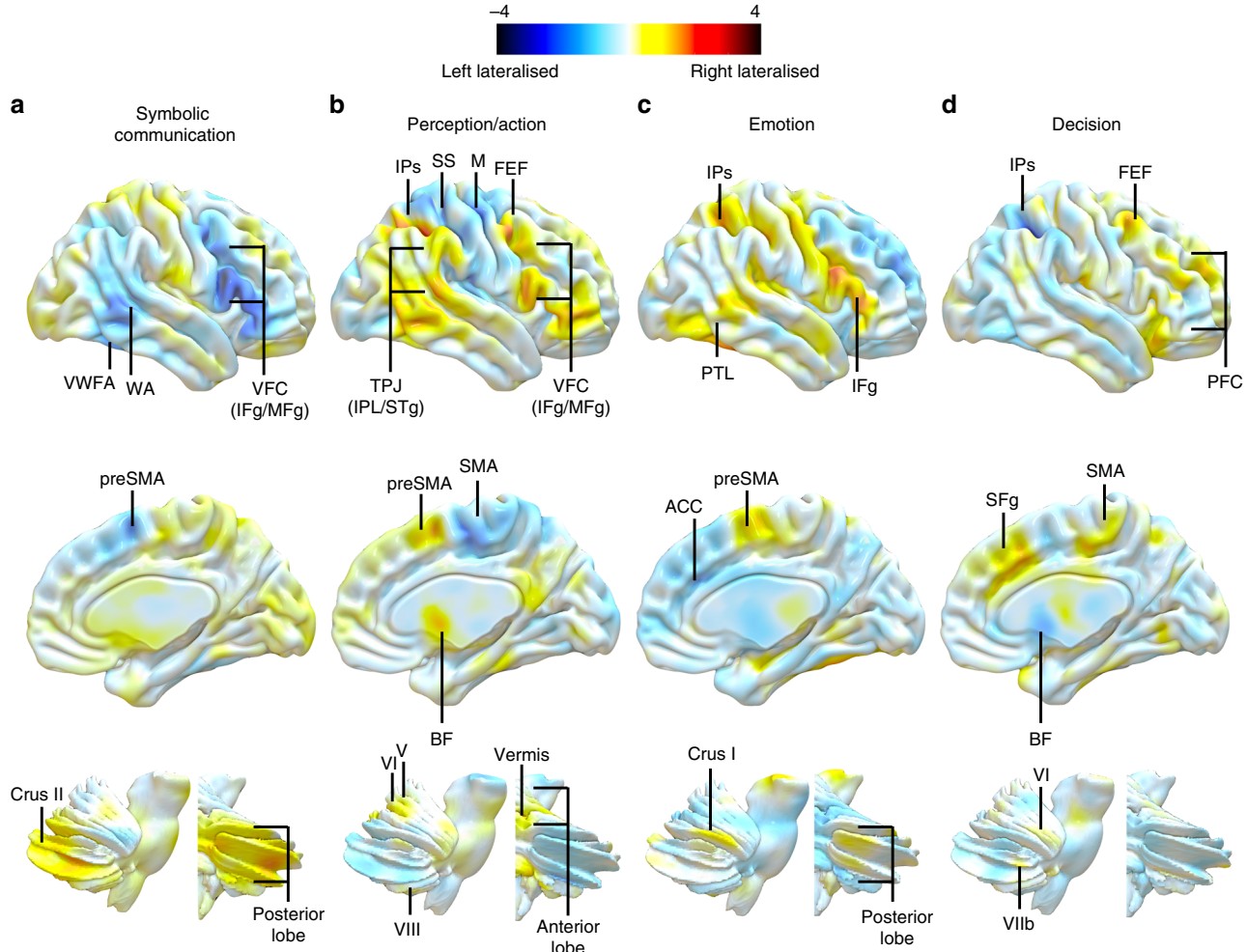

**Fig. 2** Archetypes of functional lateralisation. The maps correspond to the symbolic communication (**a**), perception/action (**b**), emotion (**c**) and decision (**d**) axes. Upper panel corresponds to the lateral view, middle panel to the medial view and lower panel to the cerebellum view (lateral and posterior views) of the reconstructed pattern of activations. VWFA visual word form area, WA Wernicke area, VFC ventral frontal cortex, IFg inferior frontal gyrus, MFg middle frontal gyrus, TPJ temporo-parietal junction, IPL inferior parietal lobule, STg superior temporal gyrus, IPs intraparietal sulcus, SS somatosensory cortex, M motor cortex, FEF frontal eye field, PTL posterior temporal lobe, PFC prefrontal cortex, SMA supplementary motor area, preSMA presupplementary motor area, ACC anterior cingulate cortex, BF basal forebrain (maps are available as Supplementary Data 2–5)

decision-making. Additionally, lateralised regions, as compared to non-lateralised regions, were connected to regions of the corpus callosum with reduced microstructural connectivity. Finally, within the pool of lateralised regions, corpus callosum macro-structural connectivity was proportionally associated with the degree of hemispheric functional dominance.

The meta-analysis of task-related activation maps in relation to cognitive terms replicated several known functional lateralisation profiles (Figs. 1 and 2). For instance, the term /language/ was associated with dominant responses in prefrontal, superior temporal regions and inferior parietal regions of the left hemisphere[32]. In association with terms such as /eye movements/stop signal/ we found several regions of right dorsal fronto-parietal and ventral frontal cortex that matched core regions of the dorsal and ventral attention network[24,33,34] involved in visuospatial and response inhibition processes. Surprisingly, the term /attention/, possibly not specific enough, was not associated with a specific lateralised component. However, it did show a strong negative weighting on the language component, corroborating previous reports of a balance between language and attention in similar brain regions[35]. Lateralised maps in the left and right cortex had

counterparts in the contralateral cerebellum in agreement with a role of the cerebellum in supporting cognition[36,37], and the known anatomical organisation of cortico-pontine-cerebellar-thalamic pathways[38]. Even the phylogenetic organisation of functional regions in cerebellar networks was respected[39]. For instance, area Crus II, part of the neo-cerebellum, connected with frontal regions involved in language, showed a significant right lateralisation for the language component. Similarly, Areas V/VI and VIIIb, an anterior superior part of the paleocerebellum connected with sensorimotor cortical regions, showed significant right lateralisation for movement and finger components. These findings support the validity and the anatomical precision of the functional lateralisation maps based on fMRI meta-analyses (also see for comparison our meta-analytic results with a task-based fMRI results on finger tapping in Supplementary Figure 4).

The overall functional lateralisation of the brain could be summarised with a low-dimensional architecture defined by spatial patterns of activity and groups of cognitive terms. This architecture defined four axes corresponding to symbolic communication, perception/action, emotion and decision-making (Fig. 1). The association of different terms along the different

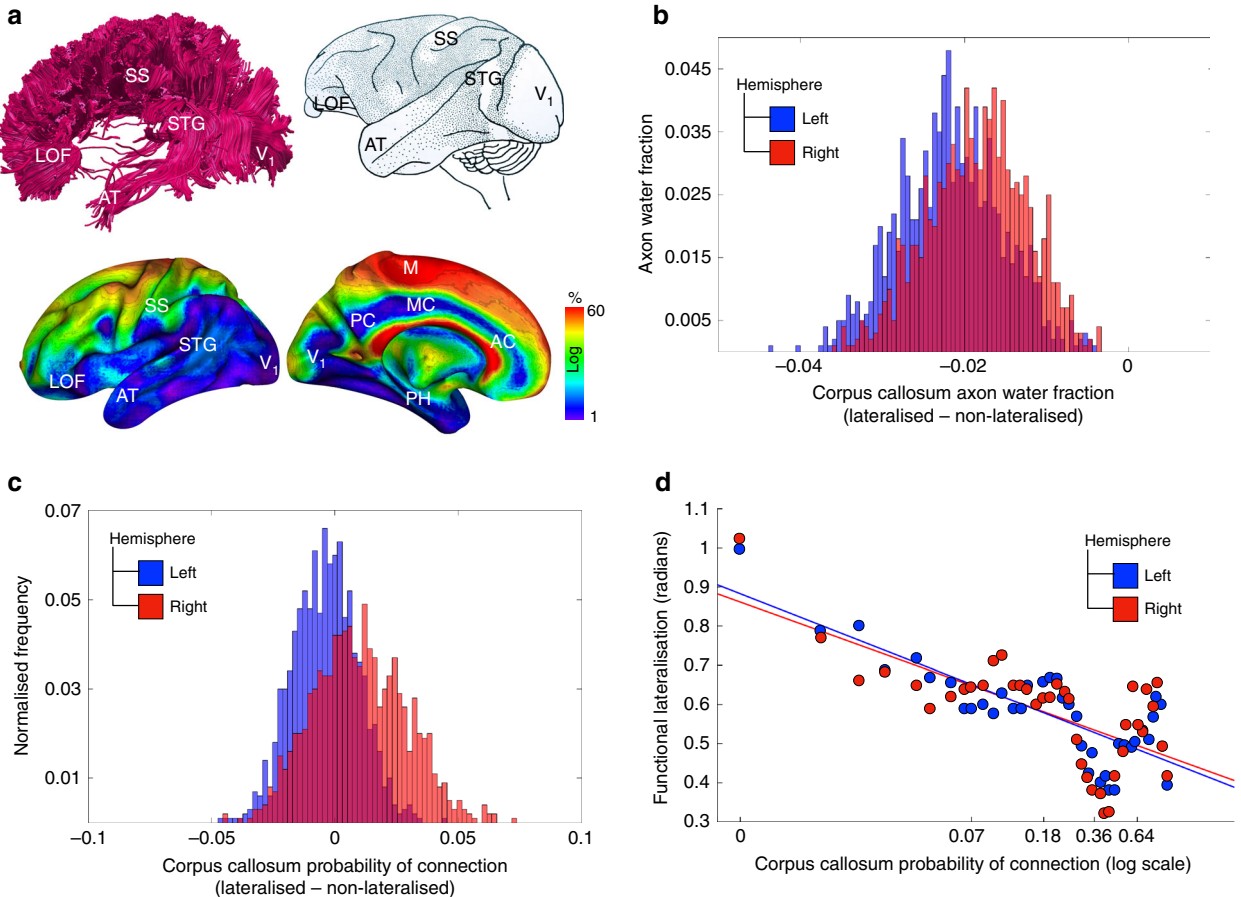

**Fig. 3** Lateralisation and inter-hemispheric connectivity. **a** Tractography of the corpus callosum in a representative subject of our study (top left); cortical projection of the corpus callosum derived from axonal tracing in monkeys[31] (top right); cortical projections of the corpus callosum derived from tractography in the participants of our study (bottom). **b** Histogram of the difference between lateralised and non-lateralised regions in the corpus callosum axonal water fraction, averaged across participants. **c** Histogram of the difference between lateralised and non-lateralised regions in the corpus callosum probability of connection. The measure was calculated as the proportion of participants in which a connection exists between brain's voxels and corpus callosum to the overall HCP sample size. **d** Dimensional relationship between the degree of functional lateralisation and the corpus callosum probability of connectivity. LOF lateral orbitofrontal cortex, SS somatosensory cortex, STG superior temporal gyrus, AT anterior temporal, V1 primary visual area, M primary motor area, PC posterior cingulate gyrus, MC middle cingulate gyrus, AC anterior cingulate gyrus, PH parahippocampal gyrus

axis defined domains of function that are not trivially associated. For instance, the axis "symbolic communication" includes not only left lateralised maps related to the term /language/ but also left and right lateralised parietal maps related to /calculation/ in agreement with recent neuropsychology[40]. The axis perception/action includes left hemisphere component related to motor planning, consistent with the effects of left lesions on motor planning (apraxia)[41,42], but also right hemisphere maps related to visuospatial attention and response inhibition. As recently shown, attention and motor deficits co-vary after focal lesions[43]. The emotion axis include right hemisphere biased maps for terms such as /expression/fearful/social interactions/, but left hemisphere foci for /autobiographical memory/.

The triangular organisation of this morphospace may be significant in relation to the theory of Pareto optimality. In evolutionary biology[27], the theory posits that in complex systems (e.g. animal morphology[27] or behaviour[44]) evolution forces trade-offs among traits: strength in one trait of high evolutionary significance, e.g. solving well one set of problems is associated with relative weakness on other problems. The trait at the vertices of the triangle represent "archetypes", that is most specialised traits. Pareto optimality distributions in human cognition and

behaviour have been recently reported in association with the ability to inhibit immediate reward for larger delayed rewards, a trait associated with numerous other cognitive, behaviour, health and socioeconomic variables[45].

The low-dimensional structure of lateralised functions is consistent with previous findings that reported a low-dimensional structure of functional networks[46] and of cognitive performance in both healthy controls[47] and patients[48]. Accordingly, individual performances or deficits are not task-specific but instead shared across a range of cognitive tasks. For example, in stroke patients, two axes of behavioural deficits, one related to language and the other to attention-motor functions, occur[43]. Our result suggests that, at least in stroke, two supplementary axes of deficits might exist along the emotional and decision-making dimensions and that these dimensions tend to be under-represented by the standard behavioural and cognitive examinations. Additionally, the similarity between the grand-scale organisation of functional lateralisation in healthy controls and behavioural deficits in stroke points to the importance of inter-hemispheric connection for recovery from stroke as shown recently by several studies[9,10]

The distribution of the probability of connection of the corpus callosum onto the brain surface matched the previous atlases that

were derived from inter-hemispheric homotopic functional connectivity analyses[49]. Extra conduction time and energy consumption are required to integrate information across hemispheres. Therefore, the role of inter-hemispheric connectivity for functional lateralisation has long been debated in the literature[50]. The current study presents a comprehensive demonstration that functional lateralisation is linked to a decrease of callosal function[51] (i.e. an inter-hemispheric independence), possibly through the mechanisms of callosal myelination and pruning[52]. The alternative hypothesis that functional lateralisation depends on a competition between the hemispheres that inhibits each other via the corpus callosum, hence predicting stronger connectivity in lateralised regions, is not supported. Notably, reduced inter-hemispheric communication may improve processing time of lateralised functions, but it may lead to a decreased capacity to recover after a brain injury. This is an issue that deserves further studies as recent studies indicate a proportional recovery similar for different functions (motor, vision, visuospatial attention, language, memory)[53].

It is important to stress that several factors limited the interpretation of the findings. For instance, while the meta-analytic approach has the power to summarise thousands of task-related fMRI findings, it is limited by publication biases which prevent a generalisation of the current findings to all brain functions[54]. Additionally, the experimental paradigms probing brain function may systematically use the same or similar material which may have biased some of the asymmetries reported. For example, processes such as emotion are frequently assessed using emotional faces that typically involve the right hemisphere more than the left hemisphere[55,56]. Out of the 300 most relevant studies for the term "emotion" in the Neurosynth database, 36% used face stimuli, 28% visual scenes, 16% language-related material, 4% movies, 4% memories, 2% odour and 10% used other materials such as music, conditioned stimuli or inkblots. This appeared to have had a limited effect on our results, because the maps driving the emotion axis did not involve the face fusiform area that is specialised in face perception[57]. However, we cannot rule out the possibility that biases in label selection by the experimenters that ran the studies housed in Neurosynth may in part affect our findings. Another issue concerns whether the left lateralisation of some functions, such as finger tapping, movement and touch, could be related to the laterality of stimulus presentation or response. While we cannot rule out this possibility, lesion studies indicate that apraxia, a deficit of motor planning and control, occurs more frequently and severely after left hemisphere damage[41,42]. Moreover, the effect of the laterality of stimulus presentation or response is often counterbalanced by the use of both hands or mask out using control tasks. For instance, a large proportion (41%) of studies associated with /finger tapping/ required responses with both hands. In addition, we found an agreement between the foci of lateralised response in left SMA and left thalamus identified in our meta-analysis, and the results of a finger tapping task in a functional MRI study of 142 right-handed participants that controlled for the laterality of the manual response (Supplementary Figure 4). A third limitation, which is not specific to the current study, is that fMRI signal on the medial wall can be blurred at the acquisition stage, due to voxel size and spatial smoothing applied to the fMRI data as a standard (and typically compulsory) preprocessing step. This problem can limit the ability to detect lateralised regions along the medial wall of the brain or in regions close to the midline. Even though we observed several lateralised regions on the medial walls of the brain, it is not possible to estimate how many putatively lateralised regions were lost due to limited spatial resolution. Finally, the limitation of the connectivity analyses derived from diffusion-weighted imaging[58] also prevented us

from investigating with confidence the distinct contribution of homotopic and heterotopic areas to the functional lateralisation as well as smaller inter-hemispheric connections such as the anterior commissure, hippocampal commissure, massa intermedia (i.e. thalamus), tectal commissure of Forel (i.e. tegmentum), habenular commissure and reticular commissure (i.e. brainstem)[59]. The advent of new diffusion imaging methods[60], as well as post-mortem investigations[61], might circumvent this bias in the future.

In conclusion, the present analysis provides us with a comprehensive view of functional lateralisation in humans, which appears to be organised in four domains: symbolic communication, perception/action, emotion-related and decision-making functions. It also reveals some of its mechanisms, such as the relationship between functional lateralisation and the strength of communication between the hemispheres. The similarity between the current findings and recent work on neurological symptoms give rise to new hypotheses on the mechanisms that support brain recovery after a brain lesion.

## Methods

**Datasets**. In this study we used a meta-analytic approach to the functional MRI studies described by Yarkoni et al.[18] (http://Neurosynth.org). We downloaded the Neurosynth database that contained 3107 reversed unthresholded functional maps and the details of 11,406 literature sources as of the 25th of September 2017.

Structural connectome data were derived from the diffusion-weighted imaging dataset of 163 participants acquired at 7 Tesla by the Human Connectome Project Team[62] (http://www.humanconnectome.org/study/hcp-young-adult/) (WU-Minn Consortium; Principal investigators: David Van Essen and Kamil Ugurbil; 1U54MH091657). This was funded by the 16 NIH Institutes and Centers that support the NIH Blueprint for Neuroscience Research, and by the McDonnell Center for Systems Neuroscience at Washington University.

**Preprocessing of Neurosynth data**. Two researchers (V.R.K. and M.T.S.) acted as judges, selecting terms that, in their view, were related to specific cognitive processes. The selection procedure consisted of two stages. During the first stage, the judges made their selection independently. Brain anatomical (e.g., "salience network"), psychiatric (e.g., "schizophrenia") and pathological (e.g., "alzheimer") terms were systematically excluded. The two judges agreed on 422 terms as related to cognitive processes as well as 2309 unrelated terms that were to be discarded (88% reproducibility). For the remaining terms, the judges made their decision together. In the end, 590 cognitive terms were selected for the study.

In the present analysis, we corrected for the anatomical differences between the left and the right hemispheres to focus on the functional asymmetries. Given that the Neurosynth functional maps are provided in the standard 2 mm MNI template space, which is not symmetric, we co-registered non-linearly the MNI template to an MNI symmetrical template, available at http://www.bic.mni.mcgill.ca/ServicesAtlases/ICBM152NLin2009, using the Greedy symmetric diffeomorphic normalisation (GreedySyN) pipeline distributed with the Advanced Normalisation Tools (ANTs, http://stnava.github.io/ANTs/)[63]. The symmetrical template was downsampled to a 2 mm voxel size to match the voxel dimensions of the standard template. The estimated transformation between non-symmetrical and symmetrical MNI spaces were then applied to all functional maps.

The following steps were performed to obtain lateralisation indices for each functional map following their co-registration with the symmetrical template. Firstly, we split the functional maps into the left- and right hemisphere parts and smoothed the resulting maps using a 6 mm FWHM Gaussian filter. We then flipped the left hemisphere maps and subtracted them from unflipped right hemisphere maps in order to obtain laterality indices (LI) maps (see[24] for a similar approach). Positive and negative values in these maps would signify a higher meta-analytic evidence for, respectively, right and left lateralisation of the function associated with a term.

**Preprocessing of structural connectome data**. The scanning parameters have previously been described in Vu et al.[62]. In brief, each diffusion-weighted imaging consisted of a total of 132 near-axial slices acquired with an acceleration factor of 3 (ref. [64]), isotropic (1.05 mm³) resolution and coverage of the whole head with a TE of 71.2 ms and with a TR of 7000 ms. At each slice location, diffusion-weighted images were acquired with 65 uniformly distributed gradients in multiple Q-space shells[65] and 6 images with no diffusion gradient applied. This acquisition was repeated four times with a $b$-value of 1000 and 2000 s mm$^{-2}$ in pairs with left-to-right and right-to-left phase-encoding directions. The default HCP preprocessing pipeline (v3.19.0) was applied to the data[66,67]. In short, the susceptibility-induced off-resonance field was estimated from pairs of images with diffusion gradient applied with distortions going in opposite directions[68] and corrected for the whole

diffusion-weighted dataset using TOPUP[69]. Subsequently, motion and geometrical distortion were corrected using the EDDY tool as implemented in FSL.

ExploreDTI toolbox for Matlab (http://www.exploredti.com[70,71]) has been used to extract estimates of axonal water fraction[28]. Next, we discarded the volumes with a b-value of 1000 s mm$^{-2}$ and whole-brain deterministic tractography was subsequently performed in the native DWI space using StarTrack software (https://www.mr-startrack.com). A damped Richardson-Lucy algorithm was applied for spherical deconvolutions[72]. A fixed fibre response corresponding to a shape factor of $\alpha = 1.5 \times 10^{-3}$ mm$^2$ s$^{-1}$ was adopted, coupled with the geometric damping parameter of 8. Two hundred algorithm iterations were run. The absolute threshold was defined as three times the spherical fibre orientation distribution (FOD) of a grey matter isotropic voxel and the relative threshold as 8% of the maximum amplitude of the FOD[73]. A modified Euler algorithm[74] was used to perform the whole-brain streamline tractography, with an angle threshold of 35°, a step size of 0.5 mm and a minimum streamline length of 15 mm.

We co-registered the structural connectome data to the standard MNI 2 mm space using the following steps: first, whole-brain streamline tractography was converted into streamline density volumes where the intensities corresponded to the number of streamlines crossing each voxel. Second, a study-specific template of streamline density volumes was generated using the Greedy symmetric diffeomorphic normalisation (GreedySyN) pipeline distributed with ANTs. This provided an average template of the streamline density volumes for all subjects. The template was then co-registered with a standard 2 mm MNI152 template using flirt tool implemented in FSL. This step produced a streamline density template in the MNI152 space. Third, individual streamline density volumes were registered to the streamline density template in the MNI152 space template and the same transformation was applied to the individual whole-brain streamline tractography using the trackmath tool distributed with the software package Tract Querier[75], and to the axonal water fraction maps, using ANTs GreedySyn. This step produced a whole-brain streamline tractography and axonal water fraction maps in the standard MNI152 space.

**Determination of functionally lateralised regions**. In these analyses, completed in two steps, we thought to identify the regions with significant functional lateralisation. See Supplementary Figure 5. In the first step, we addressed the redundancy while preserving the richness of the Neurosynth data. For instance, many selected terms were related as singular and plural forms of the same word (e.g, "visual form" and "visual forms") and therefore their maps are likely to be very similar. To this end, we reduced the dimensionality of the data using a data-driven varimax-rotated principal component (PC) analysis implemented in SPSS (SPSS, Chicago, IL) with the LI maps as inputs[76–78]. Following a standard principal component analysis, involving the eigendecomposition of the covariance matrix, 171 extracted orthogonal components with eigenvalues more than the grand average were submitted to the varimax-rotation procedure using Kaiser normalisation criterion[79], with a maximum of 1000 iterations for convergence. This accounted for 72.6% of variance in the data. The distribution of loadings along varimax-rotated principal components is typically skewed and only a few items receive large loadings. Subsequently, for the purpose of discussing the results, components were labelled according to the term(s) with the largest loadings (Supplementary Table 3).

In the second step, general linear modelling was employed to identify voxels with a significant lateralisation associated with a particular component. In this analysis, the principal components were used as a set of predictors to fit the LI maps and obtain beta maps, i.e., component spatial maps. The permutation test was performed to identify significantly lateralised regions. Given that varimax rotation may impose some correlations between the columns of the principal component matrix, we performed permutations on the rows of the unrotated matrix, subsequently applying component rotation and calculating a random map on each permutation in the same way as it was done for the real principal components. This procedure allowed us to mimic the correlational structure of the unpermuted data and provide a more robust test of significance. In order to account for multiple comparisons, the maximal statistics approach was used whereby the spatial map values for the real principal components were compared to the maximal (either positively or negatively) value across a whole random map on each permutation. Five thousand permutations were run. The voxels were considered as showing a significant lateralisation if they simultaneously satisfied two criteria: (1) their spatial map values were in 97.5% cases higher or lower than, respectively, maximal positive and negative the values obtained via permutations (i.e., $p < 0.05$, two-tailed and FWE-corrected); (2) they formed a cluster of at least 20 voxels. The second criterion was used to exclude small and possibly spurious effects observed in a small number of voxels.

**Multivariate embedding of lateralisation maps**. In order to characterise a low-dimensional structure of functional brain lateralisation, a spectral embedding of the LI maps was performed using eigendecomposition of graph normalised Laplacian of similarity matrix[80]. The method sought to uncover geometric features in the similarities between the lateralisation maps by converting these similarities into distances between lateralisation maps in the embedded space (the higher similarity between lateralisation profiles, the smaller the distance). Here we concentrated only on the variances which were accounted for by the 171 components analysed in the

present study. To this end, the LI maps were "de-noised," in a sense that they were reconstructed as the matrix product of 171 components and their spatial maps. Every element of the similarity matrix was calculated as a dot product taken for a pair of "denoised" LI maps across all voxels (i.e., an element of the similarity matrix was a sum of products of voxelwise values for a pair of maps). Negative values were zeroed to permit estimability. The embedding dimensions were ordered according to their eigenvalues, from small to large. The first non-informative dimension associated with a zero eigenvalue was dropped. In the analysis we sought to determine whether there exists a structure in a low-dimensional representation of the data, specifically data structural triangularity, and if it does, in how many dimensions this structure is preserved (for eigenvalue plot—see Supplementary Figure 6). The triangular structure was quantified as a t-ratio, i.e., a ratio between the area of the convex hull encompassing all points in embedded space and an encompassing triangle of a minimal area[27]. These values were compared to the t-ratios of random LI maps. These random maps were obtained by generating 2000 sets of 590 random maps via the permutation of the voxel order. For each set, random LI maps were calculated for each pair and then submitted to varimax analysis with the number of principal components = 171. The embedding procedure was identical to the procedure applied to non-random LI maps. The dimensional span of triangular organisation was evaluated by testing if t-ratio for non-random LI maps was greater than t-ratios of random LI maps in each two-dimensional subspace of embedding ($p < 0.05$, Bonferroni-corrected). The label for the axes was defined ad-hoc according to one or a few terms situated at the vertices of the triangle. Archetype maps were approximated using multiple regression approach. We first regressed the values in each voxel across the "denoised" LI maps onto corresponding maps' coordinates in the first 171 dimensions of the embedded space (i.e., matching the number of components used for "denoising"). This provided an estimated contribution of each embedded dimension to the lateralisation index. We then obtained the archetype maps by evaluating regression coefficients for the dimensions where the triangular structure was observed at the estimated locations of the archetypes (i.e., at the vertices of "simplex" - multidimensional triangular).

**Determination of non-lateralised regions**. In the following analyses we contrasted the connectivity profiles of lateralised regions with regions that do not show a significant lateralisation but nevertheless show a significant involvement at least in one function. The latter was identified by repeating the analyses outlined in the section "Determination of functionally lateralised regions" with the original Neurosynth functional maps as inputs. See Supplementary Figure 7. This rendered 69 components, accounting for 70.6% of variance. For closer comparability, the analysis was run in the symmetrical space and for the left and right hemispheres separately. The voxels were considered to have no significant lateralisation if they met the following criteria: (1) passed the significance threshold for at least one component and one hemisphere; (2) were non-overlapping with lateralised voxels; and (3) were homologues of the voxels meeting criteria (1) and (2) in the opposite hemisphere. A shortcut term "non-lateralised" regions was used to denominate voxels without significant lateralisation in the remaining text. This provides a conservative contrast for the lateralised regions because, by virtue of the frequentist statistical approach, the non-lateralised regions would also include voxels demonstrating a considerable lateralisation but failing to meet the statistical criteria of significance used in the study. The number of non-lateralised voxels was 3.6 times greater than the number of lateralised voxels.

**Measures of the connectivity strength for structure–function relationships**. The following steps were used for structure–function relationships. First, we combined the spatial maps of significantly lateralised voxels, irrespective of the left and right polarity of lateralisation. Second, we transformed the combined map back into the regular MNI space for a joint analysis with diffusion information using an inverse of the MNI non-symmetrical to MNI symmetrical template deformations estimated above. Finally, we projected the combined map onto the white matter boundary of the non-symmetrical MNI template in each hemisphere and subsequently selected tractography from these voxels to the corpus callosum. The same procedures were applied to the maps of non-lateralised regions.

Two measures for the strength of structural inter-hemispheric connectivity were analysed. The first, microstructural, measure referred to the axonal water fraction, averaged across participants in the HCP sample, in the voxels of corpus callosum which were hit by streamlines from selected lateralised (or non-lateralised) regions. The second, macrostructural, measure of connectivity, was defined in terms of connection replicability[30] between brain's voxels and corpus callosum, i.e., as a proportion of participants in which a connection exists between brain's voxels and corpus callosum to the overall HCP sample size. We will refer to this measure as a "probability of connection" for shortness.

**Comparison of the connectivity between lateralised and non-lateralised regions**. The comparison of connectivity between lateralised and non-lateralised regions was performed by sampling subsets of voxels (without replacement) from the pools of lateralised and non-lateralised cortical voxels. A sample from each pool was equal to 5% of the entire number of voxels in that pool (i.e., ensuring that the within-pool spatial frequency of drawn samples was equal between pools). For each

subset we calculated an average value for probability of connection and a weighted average for callosal axonal water fraction, where a weight for a voxel was given as a connection replicability between this voxel and any voxel in a sampled subset. A negative value would indicate a weaker connectivity of the lateralised voxels. The distributions of the difference in the connectivity measures between lateralised and non-lateralised cortical regions were obtained by repeating the procedure 1000 times and for each hemisphere separately.

**Analysis of hemispheric dominance**. The degree of functional hemispheric dominance was evaluated in radians as an arctangent of the ratio between the strengths of activation in two hemispheres. Pi/4 was subtracted from this value to ensure that the absolute magnitude of this value increases if the task activation is unilateral and decreases if both hemispheres demonstrate comparable levels of task activity. Given that a partial spatial overlap between lateralised regions associated with different components is possible, in the analyses we picked the dominance values associated with components that rendered the largest z-score in a particular voxel. In order to obtain robust estimate for the relationship between hemispheric dominance and the strength of inter-hemispheric connectivity, the voxels were binned by the probabilities of connection such that the smallest bin width was of the size equal to 1/163 and increased with the probability of connection (given by logspace function in Matlab). This procedure was used to partially compensate for the fact that only a very limited number of voxels had a high probability of connection to the corpus callosum, whereas the majority were characterised by small values. We also estimated the voxel's average activity between left and right hemispheres (i.e., (left + right hemisphere activity)/2) and used it as a covariate of non-interest in the analyses looking at the relationship between hemispheric dominance and other measures.

## Data availability
The dataset analysed during the current study are available at https://www. humanconnectome.org and http://www.neurosynth.org.
In addition, processed data are available on request to the corresponding authors michel. thiebaut@gmail.com and http://vyacheslav.karolis@ndcn.ox.ac.uk.

## Code availability
The code used in the following analyses is available on request from http://vyacheslav. karolis@ndcn.ox.ac.uk.

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

## Acknowledgements

We particularly thank Nathalie Tzourio-Mazoyer and her team (GIN) for useful discussion and for providing us with fMRI maps of left and right hands finger tapping. This project has received funding from the European Research Council (ERC) under the European Union's Horizon 2020 research and innovation programme (grant agreement No. 818521). We also thank Lauren Sakuma for useful discussion and edits to the manuscript. The research leading to these results received funding from the "Agence Nationale de la Recherche" [grant number ANR-13-JSV4-0001-01]. Additional financial support comes from the program "Investissements d'avenir" ANR-10-IAIHU-06 and the Fondation pour la Recherche Medicale [grant number DEQ20150331725]. M.C. was supported by NIH R01NS095741, and a strategic grant from the University of Padua. M. C. was a visiting professor at the Institut du Cerveau and Moelle Epiniere (ICM) in Paris where the research was conducted.

## Author contributions

V.R.K. implemented the methods, performed the analyses and wrote the manuscript. M. C. conceived and coordinated the study, and wrote the manuscript. M.T.S. conceived and coordinated the study, reviewed the neuroimaging data, wrote the manuscript and provided funding.

## Additional information

**Competing interests:** The authors declare no competing interests.

