## [Peer Review File · Nature Communications]

Reviewers' comments:

Reviewer #1 (Remarks to the Author):

This paper presents a unique study combining large datasets of functional activations and structural connectivity. The results are extremely interesting and will have importance for the improved understanding of the brain's functional lateralization. However, there is a significant amount of reader confusion regarding the methods, which can be addressed by editing to improve the manuscript clarity.

In the results, several acronyms are used without definition (PC, LI). This is a challenge for the reader to understand.

Line 117-118 "cortical projections of the corpus callosum
118 derived from tractography in humans" This text lacks an attribution or citation for the image, which appears to not have been created as part of the current study. Or if it was, that could be made more clear in the caption.

Figure 2: I have not heard of a "binary chart," and google did not provide any examples when I searched. These plots appear to be what is known as bar graphs, bar charts, or histograms. I recommend choosing a typical name such as these.

Figure 2. This may be later in the methods, but it is unclear to the reader if these graphs are from all voxels in one brain, or an average across all voxels in one brain then plotted across all subjects under study, etc. It would help to at least establish if this figure contains data from more than one subject and if it somehow summarizes the overall results of the study.

Line 187 "reduced inter-hemispheric communication would optimise the treatment time in the brain" It is unclear at this point what is being treated or what the treatment is. This might refer back to the previous paragraph that mentions stroke, but the current paragraph mentions no disease. Even if it is stroke, it is unclear how reduced interhemispheric connectivity would optimize a treatment or the time that the treatment takes.

In Methods, the Datasets section fails to mention how many HCP subjects were employed. In fact, the text does not mention diffusion MRI but says "structural connectivity data" which may mean some tractography processing was done before download? This should be clarified whether it was diffusion MRI data and how many subjects. Also if they were one of the unrelated datasets, as that would seem to be the reasonable choice.

The tractography section registers the streamline density to the MNI space, but not to the symmetric space in which the LI maps are located. Does this mean the LI maps were reverse transformed back into regular MNI space for joint analysis with diffusion information? At some point, the data from all modalities needed to be in the same space to compute the measures listed in the study. Please clarify.

It is confusing that ANTS is used to register DWI to MNI, but then there is an extra sentence saying track math was used for registration. I cannot understand what trackmath added or why a tool named trackmath would perform registration. Perhaps the authors mean something other than registration like averaging tract information across participants? This should all be clarified. What is the final output of each of these sections?

Line 282 "grouped the terms on the basis of their similarities" The explanation, definition, or citation of the similarity metric between words or phrases is missing here. Alternatively if the map similarity was used that should be specified and that similarity explained. I think possibly based on supplementary figure 1, that the similarity is defined in terms of the maps, but the embedding is

visualized using the words. However, this grouping is on line 282 then the embedding is described later on 315. I don't know if the grouping on line 282 relates to the embedding described later.

I do not quite understand the intuition behind the determination of non-lateralized regions. "The clusters which passed the significance threshold at least for one component and one hemisphere..." In general, a non-lateralized region would be expected to be present in both hemispheres, not only one hemisphere.

More detail is needed for the reader to fully understand the data inputs and outputs for processing. For example, were the PCA components created from the standard term maps? Not LI maps? If so what is the rationale for this approach of predicting LI using the term maps? Then how was this analysis repeated for each hemisphere separately where there is no LI?

On line 316, there is a spectral embedding of the LI maps. Are these the standard LI maps from subtraction across hemispheres, or were these modified somehow in the previous steps involving PCA?

"Firstly, the 318 functional maps values were de-noised, in a sense that it contained the values accounted for 319 by the linear combination of 171 PCs" This seems like a "functional map" was reconstructed according to its PCA representation. But this section is about LI embedding. Is the "functional map" the LI? The text seems like it can be the same thing but "functional map" was not previously used and has no lateralization in its name. I think the terminology needs clarification throughout this section.

"the elements of the similarity matrix were calculated as dot products across all voxels for all pairs of maps" This is not clear. A dot product is between one pair of voxels in one pair of maps. But there are many pairs due to many voxels and many functional maps. So was the final similarity an average? Across what? The whole map?

"A 345 connection was classified as existing if tractography seeded from a voxel in a participant's 346 brain" This sentence is not quite correct. Tractography was performed (seeded) in the whole brain, according to section B) line 248 and onward: "Whole-brain deterministic tractography was performed 253 in the native DWI space using StarTrack software." Therefore tractography seeded in many voxels has a possibility of reaching a particular voxel in the cortex. Therefore it is not true that only fibers seeded in that cortical voxel are considered for reaching the corpus callosum. The sentence on line 345 should be rewritten (and the rest of the section that discusses seeding from lateralized areas). Or, if in fact the whole-brain tractography was not used, that should be justified (as the brute force whole brain method is the most comprehensive for deterministic tractography) and the actual seeding method should be described, with rewritten methods for section B), etc. Perhaps the authors mean that tractography was selected from all lateralized regions rather than seeded?

Line 355: "The number of non-lateralised voxels was 3.6 times greater than the number of 356 lateralised voxels." This seems to belong in the earlier section about determination of lateralized voxels.

"358 B) Comparison of the connectivity between lateralised and non-lateralised regions" Was this process done for each subject under study? Was it 1000 samples per hemisphere per subject? Was the connectivity data somehow averaged across subjects first?

How many components were actually used? 171? So are there 171 sets of lateralized and non-lateralized regions?

Due in part to the confusion in the section about the PCA on terms vs maps, I am not sure how the

final "PC labels" in supplementary table 2 were generated. Regardless of how the PCA was performed, was there a person in the loop to label the PCs? Presumably this needed manual input?

Some minor grammatical issues, including the following and many more that I do not list, need to be corrected for proper understanding of the manuscript. I suggest a close read by a native speaker of English.

Line 39 should be "the contribution.. to the emergence... ariseS"

Really though, it is odd for the contribution to arise from an idea. Rather it is the reverse: the ideas seem to try to explain the contribution of the corpus, which arises due to evolution presumably. This text should be improved so the meaning of the paragraph becomes clear.

Line 44: " the hemispheres inhibit each other" (not inhibits)

Line 74: "used the same analysis to explore" not explored

Line 84: water .. diffuses (not diffuse)

Reviewer #2 (Remarks to the Author):

In this manuscript, Karolis et al. combine a meta-analytic approach with Human Connectome Project (HCP) data analysis in order to (i) examine the lateralization of fMRI BOLD activations in the human brain, and (ii) test how this relates to homotopic anatomical connectivity obtained from diffusion imaging. They perform dimensionality reduction to identify 3 to 4 "axes" along which substantial portions of functional lateralization are associated, and they identified these axes with "symbolic communication"; "emotion"; "perception/action" and "decision". The authors further tested whether greater functional lateralization was associated with greater (or less) inter hemispheric connectivity, and they observed that it was associated with less interhemispheric connectivity.

The conceptual structure of the manuscript is quite clear, but the sentence-level writing is often confusing. The manuscript is lacking in methodological detail; even with supplemental material, there is far too little detail or specification of what was performed. The general questions being addressed are important, but there appear to be (major) methodological confounds that undermine the central analyses. Overall, the meta-analytic approach has some benefits (breadth of functions assessed and large sample sizes), but these appear to be outweighed by its costs (muddiness of interpretation; confounding).

// Main Points //

/1/ The analysis identifying axes of lateralization does not distinguish between functions which are inherently lateralized (e.g. speech production) from functions that are only lateralized in relation to specific tasks (e.g. the "movement", "tapping", "motor performance" axis, for which left and right-sided movement may be lateralized but motor control itself is not a lateralized brain function). By this logic, "vision" would be considered a lateralized function because of the visual response to stimuli in contralateral space. This is an inherent weakness of the "key word" approach employed in the NeuroSynth analysis. This is not just an arbitrary conceptual issue, but directly bears on the later analyses: for speech productions, which is more genuinely a lateralized function, the arguments about homotopic connectivity (weaker vs stronger) make some sense, but for the motor system, the same arguments do not apply: if the function being assessed is "left vs right motor cortex in tapping of the left finger " then there is no reason for the same pattern to emerge as for "left IFG vs right IFG in language function" — because in the motor case "right finger tapping" is a function that also exists, but there is no corresponding right-sided function or region for language. This mixing of different kinds of lateralization undermines the logic of all subsequent

analyses.

/2/ A related concern is that the “amount of variance” being captured in a particular dimension may not necessarily reflect the variance in brain organization as much as the variation in which kinds of standard paradigms and standard keywords experimentalists prefer to use. For example, the “emotion” axis is meant to be strongly lateralized, but it is unclear how much of this effect relates to the fact that (a) many fMRI studies of emotion use facial stimuli and (b) face responses are lateralized. Thus, it could be that “emotional processing” is not nearly as lateralized as are the brain maps commonly derived from face-based experimental paradigms. If emotional sounds were used, then a different pattern could be obtained. The authors should make some attempt to validate that their findings do not derive from the somewhat arbitrary biases in which experimental paradigms are popular for investigating particular functions.

/3/ The analyses of corpus callosum (CC) anatomical connectivity suffer from a major confound: the distance of brain regions from the midline of the brain. Midline brain regions are less likely to be lateralized for a number of reasons: first, because spatial blurring bleeds BOLD signal across hemispheres along the midline, reducing sensitivity to lateralization; second, because (possibly) fewer midline functions are lateralized. In addition, CC connectivity will almost certainly vary as a function of distance from the midline, because of the differences in curvature and in length of the paths connecting homotopic regions near the midline. Therefore, the CC analyses all need to be performed in a manner that controls for the distance from the midline of the lateralized and non-lateralized voxels. In particular, the results in Figure 2D can be explained by this confounding factor.

/4/ It was very difficult to understand the rationale for selecting 3 (or 4) dimensions for the “low dimensional structure” presented in Figure 1. Typically, dimensionality reduction methods, when choosing the number of dimensions employ a cross-validation procedure to determine how much out-of-sample variance is captured, and how this varies as a function of the number of the maintained dimensions. Data relating to how much variance was captured, and how this falls off with dimensionality, was not (as far as I could tell) presented.

/5/ I was unable to understand why the “triangular organization” is an important phenomenon, or how this would provide evidence for a “Pareto front”. The authors need to extend their explanation and argumentation around this issue, and (given the large number of steps in obtaining the organization from the meta-analytic strategy) could use simulations that repeat (all of) these steps in order to demonstrate that triangular organizations do not arise generically in any dataset which contains some low-dimensional structure.

/6/ The authors should elaborate on how anatomical differences between left and right hemisphere were “adjusted”, before functional comparisons of left vs right. This is a crucial issue, because if some functions are slightly “shifted” across hemispheres (e.g. the right finger motor cortex is 1cm dorsal of the homotopic finger cortex in the left hemisphere) then it is unclear how this would influence the analysis. Would the functions show up as lateralized, even though corresponding regions appear in each hemisphere, just shifted?

// Smaller Points //

// “Given that water more likely diffuse within and along axons in the brain” — the authors could more carefully word this sentence, to be clear that they are referring to microscopic Brownian motion, rather than mesoscopic diffusion or even perfusion along axons. Diffusion imaging depends on the microscopic motion, not mesoscopic (e.g. voxel scale) flows.

// Line 131: “Furthermore, we repeated the same analysis regressing out the average level of activity in functionally lateralised areas.”

Please provide more information about how the average level of activity was computed, and why

this regression is an important control.

// When the authors compare lateralized and non-lateralized voxels it is important that they control the spatial autocorrelation (clustering / clunkiness) of the voxels to be equally distributed across the two groups. The non-lateralized voxels are providing a “null “ distribution, and it needs to share as many properties as possible in order to provide a good control. If non-lateralized voxels can be distributed randomly across the brain, while lateralized voxels are closer to one another, then the statistical properties of the two kinds of samples are not comparable. For example, the likelihood that two voxels will share a common path through the callosum depends on how close the two voxels are to one another. The non-lateralized voxels should be selected in a manner that matches the pairwise-proximity distribution of the lateralized voxels.

// The human CC projection map in Figure 2A should be augmented to show the medial surface in addition to the lateral surface.

Reviewer #3 (Remarks to the Author):

In this manuscript “Architecture of functional lateralization in the human brain”, Karolis and colleagues performed meta factor analysis on the neurosynth database to systematically analyze patterns of functional lateralization, and further correlate those results with inter-hemispheric structural connectivity from the 7T HCP data.

Overall, this is an elegant study and very enjoyable to read. The manuscript is well written, and while the methods were quite advanced the overall analytical design is compact and relatively easy to follow when consulting Supplementary Figures 4 and 5. Brain lateralization is certainly one of the oldest topics of research in our field, and a more modern, data-driven take on this topic should be of wide interest and a welcome addition to the field. Below are some suggestions to improve the clarity of the manuscript.

1. One of the key messages of this manuscript that functional lateralization has a “low dimensional structure”. This description is repeated throughout the manuscript, but in an abstract way that unfortunately lacks concrete description and examples. By examining the results, my understanding is that this statement essentially means there are roughly three to four spatial patterns of functional lateralization. That is, if you perform many different types of tasks that will result in lateralized brain activities, the way those activities are lateralized can be categorized to a low limited of spatial patterns. Could the authors put some effort to describe their take home in a more accessible form for the general neuroscience audience, as I believe is the targeted audience for Nature Communications. A related suggestion is to add some text and describing the four resulting axes, and describe the spatial pattern. Some can be already found in paragraph two of the discussion section, but I suggest more to give readers a more concrete idea of the results.
2. Another key finding is the “label” of the low dimensional structure. How did the authors come up with those labels like “Symbolic”, “Emotion”, “Perception/Action”, and “Decision Making”? As far as I can tell these were not from the neurosynth database. This is a distraction from your otherwise clean data-driven analysis. Was there an analytical procedure (instead of ad-hoc) that result in these labels? Similarly, how were the PCs labeled?
3. In the introduction, the authors discussed two hypotheses concerning the mechanisms of lateralization. Avoid of excessive inter-hemispheric conduction, or lateral inhibition. From Figure 2D. I gather your results support the first hypothesis, but these interesting points were never discussed in length in the discussion section. As a result, it is not clear what the authors think the mechanism is. Do the authors think the diffusion results suggest the lateral inhibition hypothesis is less likely and functional lateralization is a result of minimizing the cost of inter-hemisphere interaction? Please clarify.
4. I don't understand the rationale of editing the MNI template to make the anatomical template

symmetric. Assuming there are important asymmetric anatomical patterns, those will likely contribute to functional lateralization. By removing the natural occurring anatomical asymmetry won't you be missing out on important data points?

5. Please specify the amount of data (# of subjects) from the 7T HCP dataset that you analyzed.

Reviewers' comments:

Reviewer #1 (Remarks to the Author):

1) This paper presents a unique study combining large datasets of functional activations and structural connectivity. The results are extremely interesting and will have importance for the improved understanding of the brain's functional lateralization. However, there is a significant amount of reader confusion regarding the methods, which can be addressed by editing to improve the manuscript clarity.

We thank the reviewer for his/her positive assessment, and we tried in this revision to explain more clearly methods and results.

2) In the results, several acronyms are used without definition (PC, LI). This is a challenge for the reader to understand.

We now amended the manuscript accordingly and avoided acronyms to simplify the manuscript.

3) Line 117-118 "cortical projections of the corpus callosum derived from tractography in humans" This text lacks an attribution or citation for the image, which appears to not have been created as part of the current study. Or if it was, that could be made more clear in the caption.

We modified the caption accordingly

"(a) Tractography of the corpus callosum in a representative subject of our study (top left); cortical projection of the corpus callosum derived from axonal tracing in monkeys (Myers, 1965) (top right); cortical projections of the corpus callosum derived from tractography in the participants of our study (bottom)." p.8-9

4) Figure 2: I have not heard of a "binary chart," and google did not provide any examples when I searched. These plots appear to be what is known as bar graphs, bar charts, or histograms. I recommend choosing a typical name such as these.

Thank you for pointing out this typo. We used a histogram. This has been corrected in the text.

5) Figure 2. This may be later in the methods, but it is unclear to the reader if these graphs are from all voxels in one brain, or an average across all voxels in one brain then plotted across all subjects under study, etc. It would help to at least establish if this figure contains data from more than one subject and if it somehow summarizes the overall results of the study.

We now provide a detailed description of measures in the Result section:

“Two measures of connectivity were employed; both computed by averaging across participants in the HCP sample. The first measure, axonal water fraction (Fieremans, Jensen, & Helpert, 2011), is microscopic and is estimated in the midsection of the corpus callosum crossed by streamlines originating from voxels of a selected cortical region. The second measure is macroscopic and estimates the replicability of connections (Karolis et al., 2016; Thiebaut de Schotten et al., 2011) calculated as the proportion of participants in which a voxel is connected to the corpus callosum, which we will refer to as probability of connection for shortness.” p. 7

and in the Methods section:

“The comparison of connectivity between lateralised and non-lateralised regions was performed by sampling subsets of voxels (without replacement) from the pools of lateralised and non-lateralised cortical voxels. A sample from each pool was equal to 5% of the entire number of voxels in that pool (i.e., ensuring that the within-pool spatial frequency of drawn samples was equal between pools). For each subset we calculated an average value for probability of connection and a weighted average for callosal axonal water fraction, where a weight for a voxel was given as a connection replicability between this voxel and any voxel in a sampled subset. A negative value would indicate a weaker connectivity of the lateralised voxels. The distributions of the difference in the connectivity measures between lateralised and non-lateralised cortical regions were obtained by repeating the procedure 1000 times and for each hemisphere separately.” p. 19-20

6) Line 187 “reduced inter-hemispheric communication would optimise the treatment time in the brain” It is unclear at this point what is being treated or what the treatment is. This might refer back to the previous paragraph that mentions stroke, but the current paragraph mentions no disease. Even if it is stroke, it is unclear how reduced interhemispheric connectivity would optimize a treatment or the time that the treatment takes.

The text was not clear, and it has been corrected as follows:

" Notably, reduced inter-hemispheric communication may improve processing time of lateralized functions, but it may lead to a decreased capacity to recover after a brain injury. This is an issue that deserves further studies as recent studies indicate a proportional recovery similar for different functions (motor, vision, visuospatial attention, language, memory) (Ramsey et al., 2017)." p. 11

7) In Methods, the Datasets section fails to mention how many HCP subjects were employed. In fact, the text does not mention diffusion MRI but says “structural connectivity data” which may mean some tractography processing was done before download? This should be clarified whether it was diffusion MRI data and how many subjects. Also if they were one of the unrelated datasets, as that would seem to be the reasonable choice.

The text was not clear. We now clarified the methods accordingly:

"Structural connectome data were derived from the diffusion-weighted imaging dataset of 163 participants acquired at 7 Tesla by the Human Connectome Project team⁶⁰. " p. 14

"In brief, each diffusion-weighted imaging consisted of a total of 132 near-axial slices acquired with an acceleration factor of 3 (Moeller et al., 2010), isotropic (1.05 mm³) resolution and coverage of the whole head with a TE of 71.2 ms and with a TR of 7000 ms. At each slice location, diffusion-weighted images were acquired with 65 uniformly distributed gradients in multiple Q-space shells (Caruyer, Lenglet, Sapiro, & Deriche, 2013) and 6 images with no diffusion gradient applied. This acquisition was repeated 4 times with a b-value of 1000 and 2000 s.mm⁻² in pairs with left-to-right and right-to-left phase-encoding directions. The default HCP preprocessing pipeline (v3.19.0) was applied to the data (Andersson et al., 2012; Sotiropoulos et al., 2013). In short, the susceptibility-induced off-resonance field was estimated from pairs of images with diffusion gradient applied with distortions going in opposite directions (Andersson, Skare, & Ashburner, 2003) and corrected for the whole diffusion-weighted dataset using TOPUP (Smith et al., 2004). Subsequently, motion and geometrical distortion were corrected using the EDDY tool as implemented in FSL." p.15

8) The tractography section registers the streamline density to the MNI space, but not to the symmetric space in which the LI maps are located. Does this mean the LI maps

were reverse transformed back into regular MNI space for joint analysis with diffusion information? At some point, the data from all modalities needed to be in the same space to compute the measures listed in the study. Please clarify.

Yes, the LI maps were transformed back into regular MNI space for joint analysis with diffusion information. We now clarified this point in the text as follow:

"The following steps were used for structure-function relationships. First, we combined the spatial maps of significantly lateralised voxels, irrespective of the left and right polarity of lateralisation. Second, we transformed the combined map back into the regular MNI space for a joint analysis with diffusion information using an inverse of the MNI non-symmetrical to MNI symmetrical template deformations estimated above. Finally, we projected the combined map onto the white matter boundary of the non-symmetrical MNI template in each hemisphere and subsequently selected tractography from these voxels to the corpus callosum. The same procedures were applied to the maps of non-lateralised regions." p. 19

9) It is confusing that ANTS is used to register DWI to MNI, but then there is an extra sentence saying track math was used for registration. I cannot understand what trackmath added or why a tool named trackmath would perform registration. Perhaps the authors mean something other than registration like averaging tract information across participants? This should all be clarified. What is the final output of each of these sections?

This is an important question. It is challenging to align diffusion-weighted imaging with the MNI space using non-linear deformation due to methodological reasons including the requirement of the rotation of the gradient table independently for every voxel of the brain. In order to circumvent this issue, ANTs was employed to estimate the transformation to the MNI using information derived from the volumes of track density maps. Subsequently, trackmath was employed to apply these deformations directly to the 3D tractography. We now clarified the method accordingly:

"We co-registered the structural connectome data to the standard MNI 2mm space using the following steps: First, whole brain streamline tractography was converted into streamline density volumes where the intensities corresponded to the number of streamlines crossing each voxel. Second, a study-specific template of streamline density volumes was generated using the Greedy symmetric diffeomorphic normalisation (GreedySyN) pipeline distributed with ANTs. This provided an average template of the streamline density volumes for all

subjects. The template was then co-registered with a standard 2mm MNI152 template using flirt tool implemented in FSL. This step produced a streamline density template in the MNI152 space. Third, individual streamline density volumes were registered to the streamline density template in the MNI152 space template and the same transformation was applied to the individual whole brain streamline tractography using the trackmath tool distributed with the software package Tract Querier (Wassermann et al., 2016), and to the axonal water fraction maps, using ANTs GreedySyn. This step produced a whole brain streamline tractography and axonal water fraction maps in the standard MNI152 space.” p. 16

10) Line 282 “grouped the terms on the basis of their similarities”. The explanation, definition, or citation of the similarity metric between words or phrases is missing here. Alternatively if the map similarity was used that should be specified and that similarity explained. I think possibly based on supplementary figure 1, that the similarity is defined in terms of the maps, but the embedding is visualized using the words. However, this grouping is on line 282 then the embedding is described later on 315. I don’t know if the grouping on line 282 relates to the embedding described later.

Indeed the use of the word “similarity” for both PCA and embedding may be confusing. As the Reviewer points out these two complementary analyses utilised different features as inputs. The purpose of PCA was to eliminate redundancy in the data driven either by correlations or by trivial relationships (e.g., singular and plural forms of the same word, “visual form” and “visual forms”). The procedure effectively aims at the creating non-collinear predictors for GLM which would allow us to determine regions with significant lateralisation. Conversely, the purpose of spectral embedding is to characterise similarities between terms. To avoid confusion we no longer use the term “similarity” in relation to PCA as well as clarify our aims:

In Results:

“We first sought to determine which regions show a significant functional lateralisation. Given that selected terms could be correlated or related in a trivial way (e.g., singular and plural forms of the same word; “visual form” and “visual forms”), a varimax-rotated principal component analysis was run in order to eliminate redundancy in the data.” p. 4

“Next, a multivariate spectral embedding, based on the similarity between lateralisation maps, enabled us to examine a generic structure of the brain’s functional lateralisation profiles, i.e. its “morphospace” (Barthélemy, 2011; Luo, Wilson, & Hancock, 2003)” p. 4

In the Methods we change the section title which describes the PCA and subsequent GLM modelling to “Determination of functionally lateralised regions” and clarified the text:

“In these analyses, completed in two steps, we thought to identify the regions with significant functional lateralisation. In the first step, we addressed the redundancy while preserving the richness of the Neurosynth data. For instance, many selected terms were related as singular and plural forms of the same word (e.g, “visual form” and “visual forms”) and therefore their maps are likely to be very similar. To this end, we reduced the dimensionality of the data using a data-driven varimax-rotated principal component (PC) analysis implemented in SPSS (SPSS, Chicago, IL) with the LI maps as inputs (Abdi & Williams, 2010; Parlatini et al., 2017; Thiebaut de Schotten et al., 2017).” p. 16

We also clarify that Figure 1 summarises all results of the section and therefore we plotted lateralised components in place of actual term maps used in embedding:

*“Low dimensional structure of the functional lateralisation in the brain embedded in two- (a) and three- (b) dimensional space according to similarity in their lateralisation profile. See **Supplementary Figure 1 and 2** for the spatial embedding of all neurosynth terms. Here, to provide a graphical summary of all results reported in the section, we plotted the lateralised components maps, named by the highest-loading terms (**Supplementary Table 3**), in place of the actual neurosynth terms (**Supplementary Table 1**).” p. 5*

11) I do not quite understand the intuition behind the determination of non-lateralized regions. “The clusters which passed the significance threshold at least for one component and one hemisphere...” In general, a non-lateralized region would be expected to be present in both hemispheres, not only one hemisphere.

Thank you for pointing out this issue. In our manuscript we use the term “non-lateralised” regions as a shortcut term for homotopic areas that: a) show significant evidence of task activation at least in one of the hemispheres and b) do not show a significant lateralisation (determined by the analyses of the difference of activations between two hemispheres). However, statistically, it is possible to obtain a significant activation in one hemisphere without a significant difference of activity between hemispheres. Therefore, it is possible to have non-lateralised regions in which activity will be significant only on one side of the brain.

A 'stricter' definition (that indicated by the reviewer) would require significant activity in both hemisphere and no difference. However, our definition is adequate for the purpose of creating maps as null models for comparisons with lateralised regions. Moreover, our definition provides a more conservative comparison, as it is entirely possible that some of the selected regions were weakly lateralised.

We now clarified a Method section as follows:

"In the following analyses we contrasted the connectivity profiles of lateralised regions with regions that do not show a significant lateralisation but nevertheless show a significant involvement at least in one function. The latter was identified by repeating the analyses outlined in the section "Determination of functionally lateralised regions" with the original neurosynth functional maps as inputs. This rendered 69 components, accounting for 70.6% of variance. For closer comparability, the analysis was run in the symmetrical space and for the left and right hemispheres separately. The voxels were considered to have no significant lateralisation if they met the following criteria: 1) passed the significance threshold for at least one component and one hemisphere; 2) were non-overlapping with lateralised voxels; and 3) were homologues of the voxels meeting criteria 1) and 2) in the opposite hemisphere. A shortcut term "non-lateralised" regions was used to denominate voxels without significant lateralisation in the remaining text. This provides a conservative contrast for the lateralised regions because, by virtue of the frequentist statistical approach, the non-lateralised regions would also include voxels demonstrating a considerable lateralisation but failing to meet the statistical criteria of significance used in the study. The number of non-lateralised voxels was 3.6 times greater than the number of lateralised voxels." p. 18-19

12) More detail is needed for the reader to fully understand the data inputs and outputs for processing. For example, were the PCA components created from the standard term maps ? Not LI maps? If so what is the rationale for this approach of predicting LI using the term maps? Then how was this analysis repeated for each hemisphere separately where there is no LI?

We realize that the description of the inputs to the varimax-PC analyses was not clear. The PCA was run twice: LI maps (i.e., the difference between LH and RH) were used as inputs to determine lateralised voxels; the standard term maps were used as inputs to determine "non-lateralised" voxels. We rewrote a section in Methods concerning determination of lateralised voxels as follows (Please refer to our reply to the previous comment by the Reviewer regarding the amendments to the description of analyses for "non-lateralised voxels"):

“To this end, we reduced the dimensionality of the data using a data-driven varimax-rotated principal component (PC) analysis implemented in SPSS (SPSS, Chicago, IL) to which LI maps served as inputs (Abdi & Williams, 2010; Parlatini et al., 2017; Thiebaut de Schotten et al., 2017)” p. 16

We also revised substantially the figures describing the analysis pipelines (**Supplementary Figure 5** and **Supplementary Figure 7**) in order to include all relevant details on the outputs and the inputs.

13) On line 316, there is a spectral embedding of the LI maps. Are these the standard LI maps from subtraction across hemispheres, or were these modified somehow in the previous steps involving PCA? “Firstly, the functional maps values were de-noised, in a sense that it contained the values accounted for by the linear combination of 171 PCs” This seems like a “functional map” was reconstructed according to its PCA representation. But this section is about LI embedding. Is the “functional map” the LI? The text seems like it can be the same thing but “functional map” was not previously used and has no lateralization in its name. I think the terminology needs clarification throughout this section.

The input to this analysis were LI maps, sorry for the confusion. We clarified the terminology accordingly

“Here we concentrated only on the variances which were accounted for by the 171 components analysed in the present study. To this end, the LI maps were “de-noised,” in a sense that they were reconstructed as the matrix product of 171 components and their spatial maps.” p. 17-18

14) “the elements of the similarity matrix were calculated as dot products across all voxels for all pairs of maps” This is not clear. A dot product is between one pair of voxels in one pair of maps. But there are many pairs due to many voxels and many functional maps. So was the final similarity an average? Across what? The whole map?

We clarified this section:

“Every element of the similarity matrix was calculated as a dot product taken for a pair of “denoised” LI maps across all voxels (i.e., an element of the similarity matrix was a sum of products of voxelwise values for a pair of maps)” p. 18

15) “A connection was classified as existing if tractography seeded from a voxel in a participant’s brain” This sentence is not quite correct.

Tractography was performed (seeded) in the whole brain, according to section B) line 248 and onward: “Whole-brain deterministic tractography was performed in the native DWI space using StarTrack software.” Therefore tractography seeded in many voxels has a possibility of reaching a particular voxel in the cortex. Therefore it is not true that only fibers seeded in that cortical voxel are considered for reaching the corpus callosum. The sentence on line 345 should be rewritten (and the rest of the section that discusses seeding from lateralized areas). Or, if in fact the whole-brain tractography was not used, that should be justified (as the brute force whole brain method is the most comprehensive for deterministic tractography) and the actual seeding method should be described, with rewritten methods for section B), etc. Perhaps the authors mean that tractography was selected from all lateralized regions rather than seeded?

Again thank you. Indeed tractography was run in the whole brain and tractography was selected and not seeded. We modified this sentence as follows (and also amending the rest of the manuscript where necessary):

“First, we combined the spatial maps of significantly lateralised voxels, irrespective of the left and right polarity of lateralisation. Second, we transformed the combined map back into the regular MNI space for a joint analysis with diffusion information using an inverse of the MNI non-symmetrical to MNI symmetrical template deformations estimated above. Finally, we projected the combined map onto white matter boundary of non-symmetrical MNI template in each hemisphere and subsequently selected tractography from these voxels to the corpus callosum. The same procedures were applied to the maps of non-lateralised regions.” p. 19

16) Line 355: “The number of non-lateralised voxels was 3.6 times greater than the number of lateralised voxels.” This seems to belong in the earlier section about determination of lateralized voxels.

Agreed, we modified the manuscript accordingly

17) “B) Comparison of the connectivity between lateralised and non-lateralised regions” Was this process done for each subject under study? Was it 1000 samples

per hemisphere per subject? Was the connectivity data somehow averaged across subjects first?

We have re-written the parts of the Results section describing the connectivity measures to clarify this point. We have also added further details to the Method section describing sampling procedure. This was done in order to also address the Minor Comment 3 by Reviewer 2:

“The comparison of connectivity between lateralised and non-lateralised regions was performed by sampling subsets of voxels (without replacement) from the pools of lateralised and non-lateralised voxels. A sample from each pool was equal to 5% of the entire number of voxels in that pool (i.e., ensuring that the within-pool spatial frequency of drawn samples was equal between pools). For each subset we calculated an average value for probability of connection and a weighted average for callosal axonal water fraction, where a weight for a voxel was given as a connection replicability between this voxel and any voxel in a sampled subset. A negative value would indicate a weaker connectivity of the lateralised voxels. The distributions of the difference in the connectivity measures between lateralised and non-lateralised areas were obtained by repeating the procedure 1000 times and for each hemisphere separately.” p. 19

18) How many components were actually used? 171? So are there 171 sets of lateralized and non-lateralized regions?

We thank the reviewer for this comment. For consistency, we applied a set criterion to the number of PCs which is the number of components each explaining more variance than the mean variance across all components. This produced a different number of components for lateralised and non-lateralised regions (171 and 69), but explaining a similar amount of variance in the data (72.6% and 70.6%). We now provide all these numbers in the manuscript. We should note that the number of components is largely irrelevant for statistical comparisons between lateralised and non-lateralised regions, because no “per-component” analyses were performed comparing them.

19) Due in part to the confusion in the section about the PCA on terms vs maps, I am not sure how the final “PC labels” in supplementary table 2 were generated.

Regardless of how the PCA was performed, was there a person in the loop to label the PCs? Presumably this needed manual input?

The label of the PCs required a manual input, which consisted in observing the loading for each term along components. Because of the varimax-rotation performed on the set of originally determined PCs, the procedure typically results in only a few terms having a large loading. In Supplementary Materials of the revised version we provide the table with top 10 loaded terms for each significant component. We clarified the methods accordingly:

“The distribution of loadings along varimax-rotated PCs is typically skewed and only a few items receive large loadings. Subsequently, for the purpose of discussing the results, components were labelled according to the term(s) with the largest loadings (Supplementary Table 3).” p. 17

20) Some minor grammatical issues, including the following and many more that I do not list, need to be corrected for proper understanding of the manuscript. I suggest a close read by a native speaker of English.

The revised manuscript has now been proof read by two native speakers.

21) Line 39 should be “the contribution.. to the emergence... ariseS”

Really though, it is odd for the contribution to arise from an idea. Rather it is the reverse: the ideas seem to try to explain the contribution of the corpus, which arises due to evolution presumably. This text should be improved so the meaning of the paragraph becomes clear.

Thank you we now edited the text accordingly.

“Two competing hypotheses have been proposed on the emergence of functional lateralisation based on the structure-function of the corpus callosum, the most considerable inter-hemispheric connection.” p. 3

22) Line 44: “ the hemispheres inhibit each other” (not inhibits)

Done

23) Line 74: “used the same analysis to explore” not explored

Done

24) Line 84: water .. diffuses (not diffuse)

Done

Thank you.

Reviewer #2 (Remarks to the Author):

In this manuscript, Karolis et al. combine a meta-analytic approach with Human Connectome Project (HCP) data analysis in order to (i) examine the lateralization of fMRI BOLD activations in the human brain, and (ii) test how this relates to homotopic anatomical connectivity obtained from diffusion imaging. They perform dimensionality reduction to identify 3 to 4 “axes” along which substantial portions of functional lateralization are associated, and they identified these axes with “symbolic communication”; “emotion”; “perception/action” and “decision”. The authors further tested whether greater functional lateralization was associated with greater (or less) inter hemispheric connectivity, and they observed that it was associated with less interhemispheric connectivity.

The conceptual structure of the manuscript is quite clear, but the sentence-level writing is often confusing. The manuscript is lacking in methodological detail; even with supplemental material, there is far too little detail or specification of what was performed. The general questions being addressed are important, but there appear to be (major) methodological confounds that undermine the central analyses. Overall, the meta-analytic approach has some benefits (broadness of functions assessed and large sample sizes), but these appear to be outweighed by its costs (muddiness of interpretation; confounding).

Thank you for your comments. We have significantly expanded the description of the methods, and added two figures (Supplementary Figures 5 and 7) that provide a complete description of the flow of analysis. We also present additional analyses to address the Reviewer’s concerns. Please see below our point-by-point responses.

// Main Points //

/1/ The analysis identifying axes of lateralization does not distinguish between functions which are inherently lateralized (e.g. speech production) from functions that

are only lateralized in relation to specific tasks (e.g. the “movement”, “tapping”, “motor performance” axis, for which left and right-sided movement may be lateralized but motor control itself is not a lateralized brain function). By this logic, “vision” would be considered a lateralized function because of the visual response to stimuli in contralateral space. This is an inherent weakness of the “key word” approach employed in the NeuroSynth analysis. This is not just an arbitrary conceptual issue, but directly bears on the later analyses: for speech productions, which is more genuinely a lateralized function, the arguments about homotopic connectivity (weaker vs stronger) make some sense, but for the motor system, the same arguments do not apply: if the function being assessed is “left vs right motor cortex in tapping of the left finger ” then there is no reason for the same pattern to emerge as for “left IFG vs right IFG in language function” — because in the motor case “right finger tapping” is a function that also exists, but there is no corresponding right-sided function or region for language. This mixing of different kinds of lateralization undermines the logic of all subsequent analyses.

This is an important point, we understand the concern of the reviewer but partially disagree, for three reasons:

1/ Motor control is a lateralised function as demonstrated in apraxia studies in patients with brain lesions (see Goldenberg G. *Apraxia: The Cognitive Side of Motor Control*, Oxford University Press 2013). Apraxia is indeed more severe and bilateral, i.e. involves both hands, after a left hemisphere lesion and less severe and contralateral after a right hemisphere lesion. There is also a large neuroimaging literature that supports the relative left hemisphere prevalence for motor control, especially in premotor regions.

2/ We have read each study harvested in the meta-analysis on finger tapping, and these studies included right hand (39), left hand (4) and both hands (31) related activations. Our analysis of the asymmetry revealed a difference of activations outside the motor cortex in the SMA and thalamus thus suggesting that the imbalance between right hand and left hand for the finger tapping task did not lead to an asymmetrical activation of the motor cortex. The absence of significant asymmetry in the motor cortex indicates that the predominance of right-hand related activations in the original meta-analysis of finger tapping did not impact significantly the functional asymmetry results.

3/ Finally, we have carried out a direct comparison between the asymmetries reported in our meta-analysis and functional activation asymmetries previously reported in right-handed participants performing left-hand and right-hand finger tapping (Tzouriot-Mazoyer et al. 2015). As shown in the enclosed figure (now Supplementary Figure 4), a direct contrast

between right and left hand movements reveals the same asymmetries in left SMA and thalamus that we report. This correspondence between the meta-analytic work and independent direct fMRI measurements support the validity of the findings reported in our study. We have report this information in Supplementary Materials.

Supplementary figure 4: Validation of the functional asymmetries derived from the meta-analysis of functional MRI with raw functional MRI results derived from a finger tapping task in 142 right-handed participants (Tzouriot-Mazoyer et al. 2015). a) functional lateralisation map for finger tapping task derived from the meta-analytic approach b) Subtraction between left-hand and left-right flipped right-hand finger tapping symmetric functional MRI maps.

We have also acknowledged this point as a potential limitation:

“Another issue concerns whether the left lateralisation of some functions, such as finger tapping, movement and touch, could be related to the laterality of stimulus presentation or response. While we cannot rule out this possibility, lesion studies indicate that apraxia, a deficit of motor planning and control, occurs more frequently and severely after left hemisphere damage (Della Sala, Faglioni, Motto, & Spinnler, 2006; Goldenberg, 2003). In addition, we found an excellent agreement between the foci of lateralised response in left SMA and left thalamus identified in our meta-analysis, and the results of a finger tapping task in a functional MRI study of 142 right-handed participants that controlled for the laterality of the manual response (Supplementary Figure 4).” p. 12

/2/ A related concern is that the “amount of variance” being captured in a particular dimension may not necessarily reflect the variance in brain organization as much as the variation in which kinds of standard paradigms and standard keywords experimentalists prefer to use. For example, the “emotion” axis is meant to be strongly lateralized, but it is unclear how much of this effect relates to the fact that (a) many fMRI studies of emotion use facial stimuli and (b) face responses are lateralized. Thus, it could be that “emotional processing” is not nearly as lateralized as are the brain maps commonly derived from face-based experimental paradigms. If emotional sounds were used, then a different pattern could be obtained. The authors should make some attempt to validate that their findings do not derive from the somewhat arbitrary biases in which experimental paradigms are popular for investigating particular functions.

This is an excellent point. In our analysis, the emotion axis is mainly driven by the lateralisation maps associated with the labels 'emotion' and 'emotional' (supplementary figures 1 and 2). We checked the 300 most relevant studies for the term emotion and we found that 36% were using face stimuli, 28% visual scenes, 16% language related material, 4% movies, 4% memories, 2% odour and 10% used other materials such as music, conditioned stimuli or inkblots. Therefore, while emotion related studies did use faces to a large extent, faces were not used in the majority of the studies. Also, while we cannot exclude the potential influence of face processing material on the measure of emotional processes, the maps associated with the terms 'emotion' and 'emotional' did not recruit the fusiform face area that was instead found in studies that included labels such as 'face' or 'faces'. Moreover, maps associated with other labels that define the emotion axis, e.g. 'expression' or 'fearful' did not include the face fusiform area. Finally, even though 'face' and 'faces' were indeed located along the emotion axis, they were not located at the peak of the emotion axis indicating that their weight was not dominant.

In sum, our results suggest that although emotions processes are often studied using faces material, it had little influence on the emotion axis reported in the manuscript. This point is now explicitly discussed. However, since the possibility of potential biases in the denomination of experimental paradigms cannot be excluded, as correctly noted by the Reviewer, we decided to acknowledge this point in the discussion:

“Additionally, the experimental paradigms probing brain function may systematically use the same or similar material which may have biased some of the asymmetries reported. For instance, processes such as emotion are frequently assessed using emotional faces that typically involve the right hemisphere more than the left hemisphere (Davidson, Shackman, & Maxwell, 2004; Gazzaniga & Smylie, 1983). Out of the 300 most relevant studies for the

term “emotion” in the Neurosynth database, 36% used face stimuli, 28% visual scenes, 16% language-related material, 4% movies, 4% memories, 2% odour and 10% used other materials such as music, conditioned stimuli or inkblots. This, however, appeared to have had a limited effect on our results, because the maps driving the emotion axis did not involve the face fusiform area that is specialised in face perception (Kanwisher, McDermott, & Chun, 1997). However, we cannot rule out the possibility that biases in label selection by the experimenters that ran the studies housed in neurosynth may in part affect our findings” p. 12

/3/ The analyses of corpus callosum (CC) anatomical connectivity suffer from a major confound: the distance of brain regions from the midline of the brain. Midline brain regions are less likely to be lateralized for a number of reasons: first, because spatial blurring bleeds BOLD signal across hemispheres along the midline, reducing sensitivity to lateralization; second, because (possibly) fewer midline functions are lateralized. In addition, CC connectivity will almost certainly vary as a function of distance from the midline, because of the differences in curvature and in length of the paths connecting homotopic regions near the midline. Therefore, the CC analyses all need to be performed in a manner that controls for the distance from the midline of the lateralized and non-lateralized voxels. In particular, the results in Figure 2D can be explained by this confounding factor.

Firstly, we agree with the Reviewer that spatial smoothing can blur the signal across hemispheres decreasing sensitivity of the lateralisation analyses for medial regions. However, this is not a specific limitation of our study but applies to all fMRI studies analysing data in a volumetric space. Our pre-processing pipeline does not add to this issue because we apply spatial smoothing after splitting the brain into two hemispheres. We would also like to note that we found several lateralised regions located medially. Nevertheless, we acknowledge this in limitations as follows:

“A third limitation, which is not specific to the current study, is that fMRI signals on the medial wall can be blurred at the acquisition stage due to voxel size and spatial smoothing applied to the fMRI data as a standard (and typically compulsory) pre-processing step. This problem can limit the ability to detect lateralised regions along the medial wall of the brain or in regions close to the midline. Even though we observed several lateralised regions on the medial wall, it is not possible to estimate how many putatively lateralised regions were lost due to limited spatial resolution.” p. 12

Secondly, thank you for pointing at the potential influence of distance as a potential confound of the connectivity analysis. This issue was thoroughly explored in the reviewed version of the manuscript.

We plotted the relationship between the two connectivity variables studied in the manuscript and the distance from the corpus callosum, and are happy to report that there is no relationship to regress from the analysis. The result is presented in Supplementary Figure 3.

Differences in distance from the corpus callosum cannot explain the proportional relationship between the degree of lateralisation of functions and the probability of connection of the corpus callosum.

We now added this supplementary analysis to the manuscript in order to clarify the results:

"Additional supplementary analyses indicated that there was no relationship between the difference in corpus callosum connectivity of lateralised and non-lateralised voxels and their distance from the mid-section of the corpus callosum (Supplementary Figure 3)." p. 9

Supplementary figure 3: Relationship between the two connectivity variables studied in the manuscript and the distance from the midsection of the corpus callosum.

/4/ It was very difficult to understand the rationale for selecting 3 (or 4) dimensions for the “low dimensional structure” presented in Figure 1. Typically, dimensionality reduction methods, when choosing the number of dimensions employ a cross-validation procedure to determine how much out-of-sample variance is captured, and how this varies as a function of the number of the maintained dimensions. Data relating to how much variance was captured, and how this falls off with dimensionality, was not (as far as I could tell) presented.

The procedure of dimensionality reduction, suggested by the Reviewer, is relevant if one seeks for a parsimonious format for describing the data in the presence of measurement noise. However, in these analyses we do not pursue this goal. Our goal is to describe a low-dimensional structure of the data (which themselves are statistical maps, not the raw data per se), not a parsimonious way of describing the data. Because of a theoretical interest in the triangular structures, we analysed in how many dimensions this structure would be preserved.

We acknowledge that that we might have created this confusion by using inconsistent terminology, e.g., “characterise low dimensionality” whereas it should be “characterise a low-dimensional structure”. We change the manuscript to accentuate the aims and to implement consistent terminology:

In results:

“Next, a multivariate spectral embedding, based on the similarity between lateralisation maps, enabled us to examine a generic structure of the brain’s functional lateralisation profiles, its “morphospace” (Barthélemy, 2011; Luo et al., 2003).” p. 4

And in Methods:

*“In the analysis we sought to determine whether there exists a structure in a low dimensional representation of the data, specifically data structural triangularity, and if it does, in how many dimensions this structure is preserved (for eigenvalue plot - see **Supplementary Figure 6**)” p. 18*

We also added a clarification on what the method does:

“The method sought to uncover geometric features in the similarities between the lateralisation maps by converting these similarities into distances between lateralisation

maps in the embedded space (the higher similarity between lateralisation profiles, the smaller the distance). ” p. 17

Finally, we provide the plot of graph Laplacian eigenvalues as a Supplementary Figure 6. These however cannot be interpreted as “variances”. The relationship of eigenvalues to the graph properties is an area of research, which we think is not within the remit of this paper.

Supplementary Figure 6. Eigenvalues of graph Laplacian embedding

/5/ I was unable to understand why the “triangular organization” is an important phenomenon, or how this would provide evidence for a “Pareto front”. The authors need to extend their explanation and argumentation around this issue, and (given the large number of steps in obtaining the organization from the meta-analytic strategy) could use simulations that repeat (all of) these steps in order to demonstrate that triangular organizations do not arise generically in any dataset which contains some low-dimensional structure.

We clarified that the t-ratio test applied to determine the triangularity is based on the comparison of our data to simulated datasets.

” Next, a multivariate spectral embedding, based on the similarity between lateralisation maps, enabled us to examine a generic structure of the brain’s functional lateralisation profiles, its “morphospace” (Barthélemy, 2011; Luo et al., 2003). The preliminary step that

included the embedding in the first two dimensions (Figure 1a and Supplementary Figure 1) revealed a triangular organisation of the lateralisation maps with three vertices: symbolic communication, perception/action, and emotion. A t-ratio test, i.e. a test of (i.e. pareto optimality, Shoval et al., 2012), between the organisation of real data and 2000 samples of simulated data that was obtained via permutations of the voxel order before computing right-left differences, confirmed the statistical veracity of such triangular organisation. The same analysis was used to explore other dimensions and revealed three additional triangles and a 4th vertex given by decision making. (Figure 1b and Supplementary Figure 2). “ p. 4-5

We also add a clarification to the Methods:

“These random maps were obtained by generating 2000 sets of 590 random maps via the permutation of the voxel order.” p. 18

And also in Discussion:

“The triangular organization of this morphospace may be significant in relation to the theory of Pareto optimality. In evolutionary biology (Shoval et al., 2012), the theory posits that in complex systems (e.g. animal morphology (Shoval et al., 2012; Szekely, Korem, Moran, Mayo, & Alon, 2015), animal behavior (Gallagher, Bjorness, Greene, You, & Avery, 2013), cancer (Hart et al., 2015), ammonite shells (Tendler, Mayo, & Alon, 2015), bacterial and single gene expression (Korem et al., 2015; Thogersen, Morup, Damkiaer, Molin, & Jelsbak, 2013), biological circuits (Szekely et al., 2015), structure of polymorphisms (Sheftel, Szekely, Mayo, Sella, & Alon, 2018), Escherichia coli proteome (Kocillari, Fariselli, Trovato, Seno, & Maritan, 2018)) evolution forces trade-offs among traits: strength in one trait of high evolutionary significance, e.g. solving well one set of problems is associated with relative weakness on other problems. The trait at the vertices of the triangle represent ‘archetypes’, that is most specialized traits. Pareto optimality distributions in human cognition and behaviour have been recently reported in association with the ability to inhibit immediate reward for larger delayed rewards, a trait associated with numerous other cognitive, behaviour, health, and socioeconomic variables (Cona et al., 2018).” p. 9

/6/ The authors should elaborate on how anatomical differences between left and right hemisphere were “adjusted”, before functional comparisons of left vs right. This is a crucial issue, because if some functions are slightly “shifted” across hemispheres (e.g. the right finger motor cortex is 1cm dorsal of the homotopic finger cortex in the left hemisphere) then it is unclear how this would influence the analysis. Would the functions show up as lateralized, even though corresponding regions appear in each hemisphere, just shifted?

Thank you for raising this important issue. This issue has now been clarified in the Results. We agree with the Reviewer that some processes or functions, e.g. the control of fingers of the right or left hand, may be shifted, and this may reflect anatomical asymmetries, e.g. some development variable that pushes more dorsally the cortico-spinal tract or callosal fibers on one side, or functionally, e.g. the left and right fingers move in a slightly different manner hence their foci of activity are slightly displaced. Characterising lateralisation from both perspectives is problematic, because there is no one-to-one mapping between functional and structural features of brain organisation of the brain.

We do not think however that this represents a conceptual problem; in fact, by concentrating on the anatomical perspective, our study attempts to investigate (if not directly) possible anatomical underpinnings of such a shift through the analyses of the relationship between brain lateralisation and structural connectivity. Our corrections for a “shift” are performed in the anatomical “reference frame” via registering term maps to the symmetrical template. We also applied smoothing to the maps to account minimise the effect of small (trivial) anatomical discrepancies in functional loci across hemispheres.

The following are our edits to the text in Results and Methods, respectively:

*“We selected 590 terms related to specific cognitive processes out of the whole Neurosynth database (see **Supplementary Table 1**). A functional lateralisation map was computed for each term by calculating the difference between hemispheres for each pair of homologous voxels. Homologous functional regions may be displaced in the two hemispheres because of anatomical factors, e.g. the Yakovlevian torque (Toga & Thompson, 2003; Yakovlev & Rakic, 1966), the size of the planum temporale (Geschwind & Levitsky, 1968) and motor cortex (Amunts et al., 1996; Yousry et al., 1997), or functional factors, e.g. the way perception and action are coupled in each hemisphere (Berlucchi, Heron, Hyman, Rizzolatti, & Umiltà, 1971; Byblow, Chua, & Goodman, 1995). Here we adjusted for main anatomical asymmetries in the two hemispheres by registering the maps to a symmetric atlas (Shulman et al., 2010). However, putative displacements due to functional factors cannot be ruled out.” p. 4*

“In the present analysis, we corrected for the anatomical differences between the left and the right hemispheres to focus on the functional asymmetries. Given that the Neurosynth functional maps are provided in the standard 2mm MNI template space, which is not symmetric, we co-registered non-linearly the MNI template to a MNI symmetrical template, available at <http://www.bic.mni.mcgill.ca/ServicesAtlases/ICBM152NLin2009>, using the

Greedy symmetric diffeomorphic normalization (GreedySyN) pipeline distributed with the Advanced Normalization Tools (ANTs, <http://stnava.github.io/ANTs/>) (Avants et al., 2011)."
p. 14

// Smaller Points //

1) // “Given that water more likely diffuse within and along axons in the brain” — the authors could more carefully word this sentence, to be clear that they are referring to microscopic Brownian motion, rather than mesoscopic diffusion or even perfusion along axons. Diffusion imaging depends on the microscopic motion, not mesoscopic (e.g. voxel scale) flows.

Thank you. We restated as follows:

*“Given that the microscopic diffusion of water molecules in the brain is easier along rather than across axons, tractography derived from diffusion-weighted magnetic resonance imaging allows for peering into the structural organisation of brain connectivity (**Figure 2a**)”.*
p. 7

2) // Line 131: “Furthermore, we repeated the same analysis regressing out the average level of activity in functionally lateralised areas.” Please provide more information about how the average level of activity was computed, and why this regression is an important control.

By “average activity” we mean the average of LH and RH values for each component map. As this measure is non-independent of the measure of hemispheric dominance, we considered it as a possible confound for the association between probability of connection and hemispheric dominance.

In the revised version of the manuscript, we clarify this point:

“As the overall level of activation of two homotopic areas in the left and the right hemisphere may have an influence on its corpus callosum connections, we duplicated the same analysis after regressing out the left and right hemisphere average level of activity for every functionally lateralised voxel.” p. 8

We also add the description of how the measure of average activity was calculated:

“We also estimated the voxel’s average activity between left and right hemispheres (i.e., (left + right hemisphere activity) / 2) and used it as a covariate of non-interest in the analyses looking at the relationship between hemispheric dominance and other measures.” p. 20

3) // When the authors compare lateralized and non-lateralized voxels it is important that they control the spatial autocorrelation (clustering / clunkiness) of the voxels to be equally distributed across the two groups. The non-lateralized voxels are providing a “null “ distribution, and it needs to share as many properties as possible in order to provide a good control. If non-lateralized voxels can be distributed randomly across the brain, while lateralized voxels are closer to one another, then the statistical properties of the two kinds of samples are not comparable. For example, the likelihood that two voxels will share a common path through the callosum depends on how close the two voxels are to one another. The non-lateralized voxels should be selected in a manner that matches the pairwise-proximity distribution of the lateralized voxels.

Thank you for this suggestion. In the revised version we changed our sampling procedure to provide a match in the distributional properties of lateralised and non-lateralised voxels. However, after adjustments, we no longer have a support for significant differences in the macrostructural measure of connectivity, which was originally significant for the right hemisphere.

We adjusted manuscript where necessary to reflect these changes. The description of the sampling method goes as follow:

“The comparison of connectivity between lateralised and non-lateralised regions was performed by sampling subsets of voxels (without replacement) from the pools of lateralised and non-lateralised voxels. A sample from each pool was equal to 5% of the entire number of voxels in that pool (i.e., ensuring that the within-pool spatial frequency of drawn samples was equal between pools). For each subset we calculated an average value for probability of connection and a weighted average for callosal axonal water fraction, where a weight for a voxel was given as a connection replicability between this voxel and any voxel in a sampled subset. A negative value would indicate a weaker connectivity of the lateralised voxels. The distributions of the difference in the connectivity measures between lateralised and non-lateralised areas were obtained by repeating the procedure 1000 times and for each hemisphere separately.” p. 19-20

4) // The human CC projection map in Figure 2A should be augmented to show the medial surface in addition to the lateral surface.

Done. Thank you!

Reviewer #3 (Remarks to the Author):

In this manuscript “Architecture of functional lateralization in the human brain”, Karolis and colleagues performed meta factor analysis on the neurosynth database to systematically analyze patterns of functional lateralization, and further correlate those results with inter-hemispheric structural connectivity from the 7T HCP data.

Overall, this is an elegant study and very enjoyable to read. The manuscript is well written, and while the methods were quite advanced the overall analytical design is compact and relatively easy to follow when consulting Supplementary Figures 4 and 5. Brain lateralization is certainly one of the oldest topics of research in our field, and a more modern, data-driven take on this topic should be of wide interest and a welcome addition to the field. Below are some suggestions to improve the clarity of the manuscript.

Thank you for your positive comments.

1. One of the key messages of this manuscript that functional lateralization has a “low dimensional structure”. This description is repeated throughout the manuscript, but in an abstract way that unfortunately lacks concrete description and examples. By examining the results, my understanding is that this statement essentially means there are roughly three to four spatial patterns of functional lateralization. That is, if you perform many different types of tasks that will result in lateralized brain activities, the way those activities are lateralized can be categorized to a low limited of spatial patterns. Could the authors put some effort to describe their take home in a more accessible form for the general neuroscience audience, as I believe is the targeted audience for Nature Communications. A related suggestion is to add some text and describing the four resulting axes, and describe the spatial pattern. Some can be

already found in paragraph two of the discussion section, but I suggest more to give readers a more concrete idea of the results.

Yes, the manuscript indeed suffers from over-condensation and we now introduce the “low dimensionality structure” concept in the introduction and we have expanded the result section to describe the anatomical pattern of the four axes.

We now say in the introduction

“However, despite the implications of functional lateralisation theories for neurodevelopmental and psychiatric disorders (Bishop, 2013; Wexler, 1980), as well as for stroke recovery (Bartolomeo & Thiebaut de Schotten, 2016; Forkel et al., 2014; Lunven et al., 2015), a comprehensive mapping of functional lateralisation in the brain is, to our knowledge, still missing in the literature. It is also not known whether putatively lateralized cognitive functions share similar or different spatial patterns of functional activation and whether these functional activations can be categorised to a limited number of spatial patterns i.e., have a low dimensional structure.” p. 3

Additionally, we have expanded the result section and produced archetype maps to describe the anatomical pattern of the four axes.

“Furthermore, by regressing lateralisation profiles onto terms’ coordinates in the embedded space, we constructed predictions for the maps located at the coordinates of the vertices, which we will refer to as archetype maps.

The archetype maps corresponding to the symbolic communication axis was characterised by a left dominant activation of the dorsal and ventral posterior part of the frontal lobe, including Broca area and the pre-supplementary motor area, the posterior part of the temporal lobe, including Wernicke area and the Visual Word Form Area (i.e. VWFA). Right dominant activations were located in the posterior lobe of the cerebellum, including area Crus II (Figure 2a).

The archetype perception/action map essentially involved left sensorimotor cortex, left SMA, and left thalamus. Right dominant activations included frontal eye field, intraparietal region, and ventral frontal regions, frontal eye field, pre-supplementary motor area, basal forebrain and anterior cerebellum (i.e. Areas V/VI and VIII) as well as part of the vermis (Figure 2b).

The archetype emotion map involved the left anterior cingulate cortex, the basolateral complex of the right amygdala, the posterior part of the right inferior frontal gyrus, the right intraparietal sulcus and the posterior part of the right temporal lobe (Figure 2c).

Finally, the decision-making archetype map involved mostly the right prefrontal cortex (i.e. medial orbital gyrus), the right frontal eye field, the left intraparietal sulcus together with the striatum (right putamen and left caudate) and the left basal forebrain (Figure 2d).

Figure 2. Archetype maps corresponding to the symbolic communication (a) perception/action (b) emotion (c) and decision (d) axes. Upper panel corresponds to the lateral view, middle panel to the medial view and lower panel to the cerebellum view (lateral and posterior views) of the reconstructed pattern of activations. VWFA, visual word form area; WA, Wernicke area; VFC, ventral frontal cortex; IFg, inferior frontal gyrus; MFg, middle frontal gyrus; TPJ, temporo-parietal junction; IPL, inferior parietal lobule; STg, superior temporal gyrus; IPs, intraparietal sulcus; SS, somatosensory cortex; M, motor cortex; FEF, frontal eye field; PTL, posterior temporal lobe; PFC, prefrontal cortex; SMA, supplementary motor area; preSMA, presupplementary motor area; somatosensory cortex, ACC, anterior cingulate cortex; BF, basal forebrain.” p. 5-6

2. Another key findings the “label” of the low dimensional structure. How did the authors come up with those labels like “Symbolic”, “Emotion”, “Perception/Action”,

and “Decision Making”? As far as I can tell these were not from the neurosynth database. This is a distraction from your otherwise clean data-driven analysis. Was there an analytical procedure (instead of ad-hoc) that result in these labels? Similarly, how were the PCs labeled?

The label of the PCs required a manual input, which consisted in observing the loading for each term along components. Because of the varimax-rotation performed on the set of originally determined PCs, the procedure typically results in only a few terms having a large loading. In Supplementary Materials of the revised version we provide the table with top 10 loaded terms for each significant component. We clarified the methods accordingly:

“The distribution of loadings along varimax-rotated principal components is typically skewed and only a few items receive large loadings. Subsequently, for the purpose of discussing the results, components were labelled according to the term(s) with the largest loadings (Supplementary Table 3).” p. 17

A similar logic was applied when labelling axes of multidimensional embedding, even though an interpretational component is present here as well.

We clarify this in the text:

“The label for the axes was defined ad-hoc according to one or a few terms situated at the vertices of the triangle” p. 18

3. In the introduction, the authors discussed two hypotheses concerning the mechanisms of lateralization. Avoid of excessive inter-hemispheric conduction, or lateral inhibition. From Figure 2D. I gather your results support the first hypothesis, but these interesting points were never discussed in length in the discussion section. As a result, it is not clear what the authors think the mechanism is. Do the authors think the diffusion results suggest the lateral inhibition hypothesis is less likely and functional lateralization is a result of minimizing the cost of inter-hemisphere interaction? Please clarify.

Thanks, we now clarified this point in the discussion:

“The current study presents a comprehensive demonstration that functional lateralisation is linked to a decrease of callosal function (Gazzaniga, 2000) (i.e. an inter-hemispheric independence), possibly through the mechanisms of callosal myelination and pruning (Luders, Thompson, & Toga, 2010), rather than to a competition between the hemispheres that inhibits each other via the corpus callosum. Hypothetically, while the reduced inter-

hemispheric communication would optimise the processing time in the brain, it would also decrease its capacity to recover after a brain injury." p.11

4. I don't understand the rationale of editing the MNI template to make the anatomical template symmetric. Assuming there are important asymmetric anatomical patterns, those will likely contribute to functional lateralization. By removing the natural occurring anatomical asymmetry won't you be missing out on important data points?

This is an important point. Asymmetries can be anatomically centred and functionally centred (see comment above in response to Reviewer 2). We clarified our motivation and logic in the Results and Methods sections, respectively:

"Homologous functional regions may be displaced in the two hemispheres because of anatomical factors, e.g. the Yakovlevian torque (Toga & Thompson, 2003; Yakovlev & Rakic, 1966), the size of the planum temporale (Geschwind & Levitsky, 1968) and motor cortex (Amunts et al., 1996; Yousry et al., 1997), or functional factors, e.g. the way perception and action are coupled in each hemisphere (Berlucchi et al., 1971; Byblow et al., 1995). Here we adjusted for main anatomical asymmetries in the two hemispheres by registering the maps to a symmetric atlas (Shulman et al., 2010). However, putative displacements due to functional factors cannot be ruled out." p. 4

"In the present analysis, we corrected for the anatomical differences between the left and the right hemispheres to focus on the functional asymmetries. Given that the Neurosynth functional maps are provided in the standard 2mm MNI template space, which is not symmetric, we co-registered non-linearly the MNI template to a MNI symmetrical template, available at <http://www.bic.mni.mcgill.ca/ServicesAtlases/ICBM152NLin2009>, using the Greedy symmetric diffeomorphic normalization (GreedySyN) pipeline distributed with the Advanced Normalization Tools (ANTs, <http://stnava.github.io/ANTs/>) (Avants et al., 2011)." p. 14

5. Please specify the amount of data (# of subjects) from the 7T HCP dataset that you analyzed.

This has now been clarified in the text. We analysed 163 subjects.

Thank you!

References

- Abdi, H., & Williams, L. J. (2010). Principal component analysis. *WIREs Comp Stat*, 433-459.
- Amunts, K., Schlaug, G., Schleicher, A., Steinmetz, H., Dabringhaus, A., Roland, P. E., & Zilles, K. (1996). Asymmetry in the human motor cortex and handedness. *Neuroimage*, 4(3 Pt 1), 216-222. doi:10.1006/nimg.1996.0073
- Andersson, J. L., Skare, S., & Ashburner, J. (2003). How to correct susceptibility distortions in spin-echo echo-planar images: application to diffusion tensor imaging. *Neuroimage*, 20(2), 870-888. doi:10.1016/S1053-8119(03)00336-7
S1053811903003367 [pii]
- Andersson, J. L., Xu, J., Yacoub, E., Auerbach, E., Moeller, S., & Ugurbil, K. (2012). A *Comprehensive Gaussian Process Framework for Correcting Distortions and Movements in Diffusion Images*. Paper presented at the ISMRM, Melbourne, Australia.
- Avants, B. B., Tustison, N. J., Song, G., Cook, P. A., Klein, A., & Gee, J. C. (2011). A reproducible evaluation of ANTs similarity metric performance in brain image registration. *Neuroimage*, 54(3), 2033-2044. doi:10.1016/j.neuroimage.2010.09.025
S1053-8119(10)01206-1 [pii]
- Barthélemy, M. (2011). Spatial networks. *Physics Reports*, 499(1-3), 1-101.
- Bartolomeo, P., & Thiebaut de Schotten, M. (2016). Let thy left brain know what thy right brain doeth: Inter-hemispheric compensation of functional deficits after brain damage. *Neuropsychologia*. doi:10.1016/j.neuropsychologia.2016.06.016
- Berlucchi, G., Heron, W., Hyman, R., Rizzolatti, G., & Umiltà, C. (1971). Simple reaction time of ipsilateral and contralateral hand to lateralized visual stimuli. *Brain*, 94, 419-430.
- Bishop, D. V. (2013). Cerebral asymmetry and language development: cause, correlate, or consequence? *Science*, 340(6138), 1230531. doi:10.1126/science.1230531
- Byblow, W. D., Chua, R., & Goodman, D. (1995). Asymmetries in Coupling Dynamics of Perception and Action. *J Mot Behav*, 27(2), 123-137. doi:10.1080/00222895.1995.9941705
- Caruyer, E., Lenglet, C., Sapiro, G., & Deriche, R. (2013). Design of multishell sampling schemes with uniform coverage in diffusion MRI. *Magn Reson Med*, 69(6), 1534-1540. doi:10.1002/mrm.24736
- Cona, G., Kocillari, L., Palombit, A., Bertoldo, A., Maritan, A., & Corbetta, M. (2018). Archetypes of human cognition defined by time preference for reward and their brain correlates: An evolutionary trade-off approach. *Neuroimage*, 185, 322-334. doi:10.1016/j.neuroimage.2018.10.050
- Davidson, R. J., Shackman, A. J., & Maxwell, J. S. (2004). Asymmetries in face and brain related to emotion. *Trends Cogn Sci*, 8(9), 389-391. doi:10.1016/j.tics.2004.07.006
- Della Sala, S., Faglioni, P., Motto, C., & Spinnler, H. (2006). Hemisphere asymmetry for imitation of hand and finger movements, Goldenberg's hypothesis reworked. *Neuropsychologia*, 44(8), 1496-1500. doi:10.1016/j.neuropsychologia.2005.11.011
- Fieremans, E., Jensen, J. H., & Helpert, J. A. (2011). White matter characterization with diffusional kurtosis imaging. *Neuroimage*, 58(1), 177-188. doi:10.1016/j.neuroimage.2011.06.006
- Forkel, S. J., Thiebaut de Schotten, M., Dell'Acqua, F., Kalra, L., Murphy, D. G., Williams, S. C., & Catani, M. (2014). Anatomical predictors of aphasia recovery: a tractography study of bilateral perisylvian language networks. *Brain*, 137(Pt 7), 2027-2039. doi:10.1093/brain/awu113
awu113 [pii]

- Gallagher, T., Bjorness, T., Greene, R., You, Y. J., & Avery, L. (2013). The geometry of locomotive behavioral states in *C. elegans*. *PLoS One*, *8*(3), e59865. doi:10.1371/journal.pone.0059865
- Gazzaniga, M. S. (2000). Cerebral specialization and interhemispheric communication: does the corpus callosum enable the human condition? *Brain*, *123* (Pt 7), 1293-1326.
- Gazzaniga, M. S., & Smylie, C. S. (1983). Facial recognition and brain asymmetries: clues to underlying mechanisms. *Ann Neurol*, *13*(5), 536-540. doi:10.1002/ana.410130511
- Geschwind, N., & Levitsky, W. (1968). Human brain: left-right asymmetries in temporal speech region. *Science*, *161*(837), 186-187.
- Goldenberg, G. (2003). Apraxia and beyond: life and work of Hugo Liepmann. *Cortex*, *39*(3), 509-524.
- Hart, Y., Sheftel, H., Hausser, J., Szekely, P., Ben-Moshe, N. B., Korem, Y., . . . Alon, U. (2015). Inferring biological tasks using Pareto analysis of high-dimensional data. *Nature Methods*, *12*, 233–235.
- Kanwisher, N., McDermott, J., & Chun, M. M. (1997). The fusiform face area: a module in human extrastriate cortex specialized for face perception. *J Neurosci*, *17*(11), 4302-4311.
- Karolis, V. R., Froudust-Walsh, S., Brittain, P. J., Kroll, J., Ball, G., Edwards, A. D., . . . Nosarti, C. (2016). Reinforcement of the Brain's Rich-Club Architecture Following Early Neurodevelopmental Disruption Caused by Very Preterm Birth. *Cereb Cortex*, *26*(3), 1322-1335. doi:10.1093/cercor/bhv305
- Kocillari, L., Fariselli, P., Trovato, A., Seno, F., & Maritan, A. (2018). Signature of Pareto optimization in the Escherichia coli proteome. *Sci Rep*, *8*(1), 9141. doi:10.1038/s41598-018-27287-3
- Korem, Y., Szekely, P., Hart, Y., Sheftel, H., Hausser, J., Mayo, A., . . . Alon, U. (2015). Geometry of the Gene Expression Space of Individual Cells. *PLoS Comput Biol*, *11*(7), e1004224. doi:10.1371/journal.pcbi.1004224
- Luders, E., Thompson, P. M., & Toga, A. W. (2010). The development of the corpus callosum in the healthy human brain. *J Neurosci*, *30*(33), 10985-10990. doi:10.1523/JNEUROSCI.5122-09.2010
- Lunven, M., Thiebaut De Schotten, M., Boursillon, C., Duret, C., Migliaccio, R., Rode, G., & Bartolomeo, P. (2015). White matter lesional predictors of chronic visual neglect: a longitudinal study. *Brain*, *138*(Pt 3), 746-760. doi:10.1093/brain/awu389
- Luo, B., Wilson, R. C., & Hancock, E. R. (2003). Spectral embedding of graphs. *Pattern Recognition*, *36*(10), 2213-2230.
- Moeller, S., Yacoub, E., Olman, C. A., Auerbach, E., Strupp, J., Harel, N., & Ugurbil, K. (2010). Multiband multislice GE-EPI at 7 tesla, with 16-fold acceleration using partial parallel imaging with application to high spatial and temporal whole-brain fMRI. *Magn Reson Med*, *63*(5), 1144-1153. doi:10.1002/mrm.22361
- Myers, R. E. (1965). Organization of forebrain commissures. In E. G. Ettlinger (Ed.), *Functions of the Corpus Callosum* (pp. 133–143). London: CIBA Foundation Study Group 20.
- Parlatini, V., Radua, J., Dell'Acqua, F., Leslie, A., Simmons, A., Murphy, D. G., . . . Thiebaut de Schotten, M. (2017). Functional segregation and integration within fronto-parietal networks. *Neuroimage*, *146*, 367-375. doi:10.1016/j.neuroimage.2016.08.031
- Ramsey, L. E., Siegel, J. S., Lang, C. E., Strube, M., Shulman, G. L., & Corbetta, M. (2017). Behavioural clusters and predictors of performance during recovery from stroke. *Nat Hum Behav*, *1*. doi:10.1038/s41562-016-0038
- Sheftel, H., Szekely, P., Mayo, A., Sella, G., & Alon, U. (2018). Evolutionary trade-offs and the structure of polymorphisms. *Philos Trans R Soc Lond B Biol Sci*, *373*(1747). doi:10.1098/rstb.2017.0105
- Shoval, O., Sheftel, H., Shinar, G., Hart, Y., Ramote, O., Mayo, A., . . . Alon, U. (2012). Evolutionary trade-offs, Pareto optimality, and the geometry of phenotype space. *Science*, *336*(6085), 1157-1160. doi:10.1126/science.1217405

- Shulman, G. L., Pope, D. L., Astafiev, S. V., McAvoy, M. P., Snyder, A. Z., & Corbetta, M. (2010). Right hemisphere dominance during spatial selective attention and target detection occurs outside the dorsal frontoparietal network. *Journal of Neuroscience*, *30*(10), 3640-3651. doi:30/10/3640 [pii] 10.1523/JNEUROSCI.4085-09.2010
- Smith, S. M., Jenkinson, M., Woolrich, M. W., Beckmann, C. F., Behrens, T. E. J., Johansen-Berg, H., . . . Matthews, P. M. (2004). Advances in functional and structural MR image analysis and implementation as FSL. *Neuroimage*, *23*(S1), 208-219.
- Sotiropoulos, S. N., Jbabdi, S., Xu, J., Andersson, J. L., Moeller, S., Auerbach, E. J., . . . Behrens, T. E. (2013). Advances in diffusion MRI acquisition and processing in the Human Connectome Project. *Neuroimage*, *80*, 125-143.
- Szekely, P., Korem, Y., Moran, U., Mayo, A., & Alon, U. (2015). The Mass-Longevity Triangle: Pareto Optimality and the Geometry of Life-History Trait Space. *PLoS Comput Biol*, *11*(10), e1004524. doi:10.1371/journal.pcbi.1004524
- Tendler, A., Mayo, A., & Alon, U. (2015). Evolutionary tradeoffs, Pareto optimality and the morphology of ammonite shells. *BMC Syst Biol*, *9*, 12. doi:10.1186/s12918-015-0149-z
- Thiebaut de Schotten, M., ffytche, D. H., Bizzi, A., Dell'Acqua, F., Allin, M., Walshe, M., . . . Catani, M. (2011). Atlasing location, asymmetry and inter-subject variability of white matter tracts in the human brain with MR diffusion tractography. *Neuroimage*, *54*(1), 49-59. doi:10.1016/j.neuroimage.2010.07.055
- Thiebaut de Schotten, M., Urbanski, M., Batrancourt, B., Levy, R., Dubois, B., Cerliani, L., & Volle, E. (2017). Rostro-caudal Architecture of the Frontal Lobes in Humans. *Cereb Cortex*, *27*(8), 4033-4047. doi:10.1093/cercor/bhw215
- Thogersen, J. C., Morup, M., Damkiaer, S., Molin, S., & Jelsbak, L. (2013). Archetypal analysis of diverse *Pseudomonas aeruginosa* transcriptomes reveals adaptation in cystic fibrosis airways. *BMC Bioinformatics*, *14*, 279. doi:10.1186/1471-2105-14-279
- Toga, A. W., & Thompson, P. M. (2003). Mapping brain asymmetry. *Nat Rev Neurosci*, *4*(1), 37-48.
- Wassermann, D., Makris, N., Rathi, Y., Shenton, M., Kikinis, R., Kubicki, M., & Westin, C. F. (2016). The white matter query language: a novel approach for describing human white matter anatomy. *Brain Struct Funct*, *221*(9), 4705-4721. doi:10.1007/s00429-015-1179-4
- Wexler, B. E. (1980). Cerebral laterality and psychiatry: a review of the literature. *Am J Psychiatry*, *137*(3), 279-291. doi:10.1176/ajp.137.3.279
- Yakovlev, P. J., & Rakic, P. (1966). Pattern of decussation of bulbar pyramids and distribution of pyramidal tracts on two sides of the spinal cord. *Trans Amer Neurol Assoc*, *91*, 366-367.
- Yousry, T. A., Schmid, U. D., Alkadhi, H., Schmidt, D., Peraud, A., Buettner, A., & Winkler, P. (1997). Localization of the motor hand area to a knob on the precentral gyrus. A new landmark. *Brain*, *120* (Pt 1), 141-157.

Reviewers' comments:

Reviewer #1 (Remarks to the Author):

The paper is clearly improved and the authors have addressed my comments.

Reviewer #2 (Remarks to the Author):

I thank the authors for their attentive revisions.

In my first review, my summary of the manuscript was:

"The general questions being addressed are important, but there appear to be (major) methodological confounds that undermine the central analyses. Overall, the meta-analytic approach has some benefits (broadness of functions assessed and large sample sizes), but these appear to be outweighed by its costs (muddiness of interpretation; confounding)."

The authors seem to agree with me that their methods have an uncertain interpretation and are subject to confounds that are difficult to handle. They have added a number of caveats to their manuscript text to note these cases.

More specifically:

the analysis methods used in this manuscript cannot distinguish between "lateralization of brain function" and "lateralization in the arbitrary selection of tasks that researchers typically use to measure brain function". For example, finger-tapping is not tested equally often with left and right fingers. So the neutral-sounding phrase "finger tapping" is actually used to index a highly lateralized task.

There is some overlap between the motor map obtained in this meta-analysis and in a finger-tapping map obtained in a balanced study conducted by Tzourio-Mazoyer et al. (2016). However, this does not address my concern. First, the quantity and specificity of the overlap is not quantified (e.g. using DICE coefficient). What about the areas of non-overlap? How should these be interpreted? It is not enough that the method is (somewhat) successful in some cases -- the problem is that we cannot know when these problems are arising.

Related to the following comment from the authors:

"Our analysis of the asymmetry revealed a difference of activations outside the motor cortex in the SMA and thalamus thus suggesting that the imbalance between right hand and left hand for the finger tapping task did not lead to an asymmetrical activation of the motor cortex."

Can the authors explain why their method is not detecting an asymmetrical activation of the motor cortex in this case? If 39 of the studies are right-hand tapping and only 4 are left-hand, then shouldn't there be an imbalance in the left vs. right motor activation across these tasks?

I still feel that that we do not know what the central dependent variable of this analysis is measuring, and the problems of interpretation remain extremely serious. At the same time, I am not really sure what more the authors can do to remedy the uncertainty of interpretation. I see my role as a reviewer to provide constructive comments, and not to act as a "gatekeeper". So if the other reviewers are satisfied, then I won't attempt to hold up this paper.

Reviewer #3 (Remarks to the Author):

The authors did a good job responding to my comments. I have only one comment: the color choice for figure 2 is not ideal. Rainbow is not a good color palette for diverging scales. It mixes green and red and puts yellow in the middle. 5% of the scientific workforce is colorblind. Please avoid it. There are many online resources for picking better colors.

Reviewer #1 (Remarks to the Author):

The paper is clearly improved and the authors have addressed my comments.

Thank you.

Reviewer #2 (Remarks to the Author):

I thank the authors for their attentive revisions.

In my first review, my summary of the manuscript was:

"The general questions being addressed are important, but there appear to be (major) methodological confounds that undermine the central analyses. Overall, the meta-analytic approach has some benefits (breadth of functions assessed and large sample sizes), but these appear to be outweighed by its costs (muddiness of interpretation; confounding)."

The authors seem to agree with me that their methods have an uncertain interpretation and are subject to confounds that are difficult to handle. They have added a number of caveats to their manuscript text to note these cases. More specifically: the analysis methods used in this manuscript cannot distinguish between "lateralization of brain function" and "lateralization in the arbitrary selection of tasks that researchers typically use to measure brain function". For example, finger-tapping is not tested equally often with left and right fingers. So the neutral-sounding phrase "finger tapping" is actually used to index a highly lateralized task.

We would like to note that the issues the Reviewer is referring to are potentially relevant for a limited set of motor and sensory tasks, and we explicitly referred to these issues in the previous revision, provided additional data addressing them and acknowledged the limitations. More generally the distinction made by the reviewer between "lateralization of brain function" and "lateralization in the arbitrary selection of tasks that researchers typically use to measure brain function" cannot be answered if not in relative terms. Scientists must simplify in order to study complex systems. For instance, most of what we know on language comes from processing of single words. Nobody would claim that single word processing is an accurate test of language processing, yet we have learned an awful lot about language studying single words. In neuroimaging the same can be said about many of the experimental tasks developed to study different cognitive processes. In fact most of what we know about human cognitive mechanisms comes from the very artificial situation in which young college students lay flat on their back while pushing buttons to stimuli. Again, a highly artificial and simplified set of conditions from which however a lot has been learned. Therefore while the reviewer is correct in absolute terms what we present is a reasonable and innovative approach to the issue of lateralization based on the data that are currently available.

There is some overlap between the motor map obtained in this meta-analysis and in a finger-tapping map obtained in a balanced study conducted by Tzourio-Mazoyer et al. (2016). However, this does not address my concern. First, the quantity and specificity of the overlap is not quantified (e.g. using DICE coefficient). What about the areas of non-overlap? How should these be interpreted? It is not enough that the method is (somewhat) successful in some cases -- the problem is that we cannot know when these problems are arising.

The purpose of the comparison between metanalytic maps and ground truth fMRI data was to confirm that the areas reported as significantly lateralised were also significant when compared to the fMRI data (>100 participants using left and right hand). In our work we are comparing whether any voxel is more frequently reported as activated in one hemisphere compared to the other, while fMRI studies directly compare the level of activation between the left and the right hemisphere. In the latter case, some areas can be functionally asymmetric because they are more strongly activated in one hemisphere compared to the other. This difference depends on the threshold used to assess statistical significance which in turn will affect the spatial extent of the activation hence the overlap with the meta-analysis map. The issue of overlap vis-a-vis activation threshold is naughty, and it would not provide more information than just qualitatively observing that the regions are approximately the same. In response to the Reviewer's comment, we removed the term "excellent" from the characterisation of the agreement between fMRI and meta-analytical data. We also acknowledge that the agreement between the two methods is not quantified.

Related to the following comment from the authors:

"Our analysis of the asymmetry revealed a difference of activations outside the motor cortex in the SMA and thalamus thus suggesting that the imbalance between right hand and left hand for the finger tapping task did not lead to an asymmetrical activation of the motor cortex." Can the authors explain why their method is not detecting an asymmetrical activation of the motor cortex in this case? If 39 of the studies are right-hand tapping and only 4 are left-hand, then shouldn't there be an imbalance in the left vs. right motor activation across these tasks?

We would like to bring to the attention of the Reviewer that there were also 31 studies, where participants used both hands; therefore a compound effect of the laterality of the response on the meta-analytical data appear to be insufficient to pass the statistical threshold used in the current study.

Additionally, a quick eye-balling of the finger tapping meta-analysis in neurosynth clearly show a bilateral activation of the motor cortex for finger tapping

(<http://www.neurosynth.org/analyses/terms/finger%20tapping/>)

We amended the manuscript to include:

"Moreover, the effect of the laterality of stimulus presentation or response is often counterbalanced by the use of both hands or mask out using control tasks. For instance, a large proportion (41%) of studies associated with /finger tapping/ required responses with both hands."

I still feel that that we do not know what the central dependent variable of this analysis is measuring, and the problems of interpretation remain extremely serious. At the same time, I am not really sure what more the authors can do to remedy the uncertainty of interpretation. I see my role as a reviewer to provide constructive comments, and not to act as a "gatekeeper". So if the other reviewers are satisfied, then I won't attempt to hold up this paper.

Thank you for the encouraging comment

Reviewer #3 (Remarks to the Author):

The authors did a good job responding to my comments. I have only one comment: the color choice for figure 2 is not ideal. Rainbow is not a good color palette for diverging scales. It mixes green and red and puts yellow in the middle. 5% of the scientific workforce is colorblind. Please avoid it. There are many only resources for picking better colors.

This is a good point. We changed the palette for ACTC which uses different contrasts for each colour and ran some test with colourblind colleagues. Happy to modify this figure further if required. Further we made the maps available online for further inspection (supplementary datasets 1-4)

REVIEWERS' COMMENTS:

Reviewer #2 (Remarks to the Author):

I have no further comments.

Reviewer #3 (Remarks to the Author):

I hate to be a stickler on the color issue, but the color bar for figure 2 remains suboptimal and is what we call a circular color scheme, which I believe is "jet". It is not ideal for continuous diverging measurements like left to right lateralization. Please see:

<https://jakevdp.github.io/blog/2014/10/16/how-bad-is-your-colormap/>

I suggest you pick one of the "diverging" colormaps here. It is better to have only two continuous hues:

https://matplotlib.org/examples/color/colormaps_reference.html

<https://betterfigures.org/2015/06/23/picking-a-colour-scale-for-scientific-graphics/>

Reviewer #2 (Remarks to the Author):

I have no further comments.

Thank you

Reviewer #3 (Remarks to the Author):

I hate to be a stickler on the color issue, but the color bar for figure 2 remains suboptimal and is what we call a circular color scheme, which I believe is “jet”. It is not ideal for continuous diverging measurements like left to right lateralization. Please see:

<https://jakevdp.github.io/blog/2014/10/16/how-bad-is-your-colormap/>

I suggest you pick one of the “diverging” colormaps here. It is better to have only two continuous hues:

https://matplotlib.org/examples/color/colormaps_reference.html

<https://betterfigures.org/2015/06/23/picking-a-colour-scale-for-scientific-graphics/>

Done! thank you